# Antinociceptive modulation by the adhesion GPCR CIRL promotes mechanosensory signal discrimination

**Sven Dannhäuser[1,2], Thomas J Lux[3], Chun Hu[4], Mareike Selcho[1,2], Jeremy T-C Chen[3], Nadine Ehmann[1,2], Divya Sachidanandan[1,2], Sarah Stopp[1,2], Dennis Pauls[1,2], Matthias Pawlak[5], Tobias Langenhan[6], Peter Soba[4], Heike L Rittner[3]\*, Robert J Kittel[1,2]\***

[1]Department of Animal Physiology, Institute of Biology, Leipzig University, Leipzig, Germany; [2]Carl-Ludwig-Institute for Physiology, Leipzig University, Leipzig, Germany; [3]Center for Interdisciplinary Pain Medicine, Department of Anaesthesiology, University Hospital Würzburg, Würzburg, Germany; [4]Neuronal Patterning and Connectivity, Center for Molecular Neurobiology, University Medical Center Hamburg-Eppendorf, Hamburg, Germany; [5]Department of Neurophysiology, Institute of Physiology, University of Würzburg, Würzburg, Germany; [6]Rudolf Schönheimer Institute of Biochemistry, Division of General Biochemistry, Medical Faculty, Leipzig University, Leipzig, Germany

**\*For correspondence:**
rittner_h@ukw.de (HLR);
rjkittel@me.com (RJK)

**Competing interests:** The authors declare that no competing interests exist.

**Abstract** Adhesion-type GPCRs (aGPCRs) participate in a vast range of physiological processes. Their frequent association with mechanosensitive functions suggests that processing of mechanical stimuli may be a common feature of this receptor family. Previously, we reported that the *Drosophila* aGPCR CIRL sensitizes sensory responses to gentle touch and sound by amplifying signal transduction in low-threshold mechanoreceptors (Scholz et al., 2017). Here, we show that *Cirl* is also expressed in high-threshold mechanical nociceptors where it adjusts nocifensive behaviour under physiological and pathological conditions. Optogenetic in vivo experiments indicate that CIRL lowers cAMP levels in both mechanosensory submodalities. However, contrasting its role in touch-sensitive neurons, CIRL dampens the response of nociceptors to mechanical stimulation. Consistent with this finding, rat nociceptors display decreased *Cirl1* expression during allodynia. Thus, cAMP-downregulation by CIRL exerts opposing effects on low-threshold mechanosensors and high-threshold nociceptors. This intriguing bipolar action facilitates the separation of mechanosensory signals carrying different physiological information.

## Introduction

Mechanical forces are detected and processed by the somatosensory system. Mechanosensation encompasses the distinct submodalities of touch, proprioception, and mechanical nociception. Touch plays an important discriminative role and contributes to social interactions (*Abraira and Ginty, 2013*; *McGlone et al., 2014*). Nociception reports incipient or potential tissue damage. It triggers protective behaviours and can give rise to pain sensations (*Basbaum et al., 2009*). Thus, physically similar signals can carry fundamentally different physiological information, depending on stimulus intensity. Whereas innocuous touch sensations rely on low-threshold mechanosensory neurons, noxious mechanical stimuli activate high-threshold mechanosensory neurons, i.e. nociceptors. While mechanisms to differentiate these mechanosensory submodalities are essential for survival, little is known how this is achieved at cellular and molecular levels.

The activity of nociceptors can be increased through sensitization, e.g. upon inflammation, and decreased through antinociceptive processes, leading to pain relief. In both cases, G protein-coupled receptors (GPCRs) play an important modulatory role. Receptors that couple to heterotrimeric $G_{q/11}$ or $G_s$ proteins, like the prostaglandin EP2 receptor, increase the excitability of nociceptors by activating phospholipase C and adenylyl cyclase pathways, respectively. In contrast, $G_{i/o}$-coupled receptors, which are gated by soluble ligands like morphine and endogenous opioid neuropeptides generally inhibit nociceptor signalling. In mammalian nociceptors, cell surface receptors that couple to $G_{i/o}$ proteins are located at presynaptic sites in the dorsal horn of the spinal cord, where they reduce glutamate release, at somata in dorsal root ganglia (DRG), and at peripheral processes, where they suppress receptor potential generation (*Yudin and Rohacs, 2018*).

Research on mechanosensation has focussed mainly on receptors that transduce mechanical force into electrical current, and the function of such mechanosensing ion channels is currently the subject of detailed investigations. In contrast, evidence for mechano-metabotropic signal transfer and compelling models of force conversion into an intracellular second messenger response are limited, despite the vital role of metabotropic modulation in all corners of physiology (*Mederos y Schnitzler et al., 2008*; *Hoffman et al., 2011*). Adhesion-type GPCRs (aGPCRs), a large molecule family with over 30 members in humans, operate in diverse physiological settings. Correspondingly, these receptors are associated with diverse human diseases, such as developmental disorders, defects of the nervous system, allergies and cancer (*Hamann et al., 2015*; *Scholz et al., 2016*). In contrast to most members of the GPCR phylum, aGPCRs are not activated by soluble ligands. Instead, aGPCRs interact with partner molecules tethered to membranes or fixed to the extracellular matrix via their long, adhesive N-terminal domain. This arrangement positions aGPCRs as metabotropic mechanosensors, which translate a relative displacement of the receptor-bearing cell into an intracellular second messenger signal (*Langenhan et al., 2016*).

CIRL (ADGRL/Latrophilin, Lphn), one of the oldest members of the aGPCR family, is expressed in the neuronal dendrites and cilia of *Drosophila* larval chordotonal organs (ChOs), mechanosensory structures that respond to gentle touch, sound, and proprioceptive input (*Kernan, 2007*; *Scholz et al., 2015*). Here, mechanical stimulation of CIRL triggers a conformational change of the protein and activation via its tethered agonist (*Stachel*) (*Liebscher et al., 2014*; *Stoveken et al., 2015*). Signalling by the activated receptor reduces intracellular cAMP levels, which in turn sensitizes ChO neurons and potentiates the mechanically-evoked receptor potential (*Scholz et al., 2017*). In the current study, we show that CIRL also adjusts the activity of nociceptors, which respond to strong mechanical stimuli. Here, too, its function is consistent with $G_{i/o}$ coupling. However, in contrast to touch-sensitive ChO neurons, nociceptors are sensitized by elevated cAMP concentrations and toned down by an antinociceptive and *Stachel*-independent action of CIRL. As a result of curtailing cAMP production, CIRL modulates neural processing of noxious harsh and innocuous gentle touch bidirectionally. This elegant signalling logic serves signal discrimination by helping to separate mechanosensory submodalities.

## Results

*Drosophila* larvae possess two major types of peripheral sensory neurons. Monociliated type one neurons, including ChOs and external sensory organs, and type two neurons with multiple dendritic (md) projections, classified as tracheal dendrite (md-td), bipolar dendrite (md-bd), and dendritic arborization (md-da). Md-da neurons are further subdivided into four classes: C1da-C4da (*Ghysen et al., 1986*; *Bodmer and Jan, 1987*; *Grueber et al., 2002*). Previous work demonstrated prominent expression of *Cirl* in ChOs, where it modulates sensory processing of innocuous mechanical stimuli (*Scholz et al., 2015*; *Scholz et al., 2017*). In addition, *Cirl* transcription was also noted in md neurons. Motivated by this observation, we now turned our attention to C4da neurons: polymodal nociceptors, which respond to noxious temperatures, intense light and, importantly, harsh mechanical touch (*Tracey et al., 2003*; *Xiang et al., 2010*; *Zhong et al., 2010*; *Zhong et al., 2012*; *Kim et al., 2012*; *Kim and Johnson, 2014*). Degenerin/epithelial sodium channels (DEG/ENaCs) contribute to nociceptive mechanotransduction in invertebrates and mammals (*Price et al., 2001*; *Chatzigeorgiou et al., 2010*; *Zhong et al., 2010*; *Gorczyca et al., 2014*; *Guo et al., 2014*;

*Mauthner et al., 2014*). Placing a fluorescent reporter under transcriptional control of the DEG/ENaC subunit *pickpocket* (*ppk*) reliably marks C4da neurons (*Grueber et al., 2003*; *Han et al., 2011*). We therefore combined a direct *GFP-ppk* promoter fusion (*ppk-CD4::tdGFP*) with binary expression of the red photoprotein *Tomato* by a *Cirl* promoter element in the *UAS/GAL4* system (*dCirlp^{GAL4} >UAS-CD4::tdTomato*) (*Brand and Perrimon, 1993*; *Scholz et al., 2015*). This setting revealed *Cirl* transcription in both *ppk*-negative ChOs and *ppk*-positive C4da neurons (*Figure 1*). Thus, *Cirl* is expressed in different sensory neurons, including proprioceptors and nociceptors.

Given CIRL's role in mechanosensation, we next tested for a specific contribution of the aGPCR to mechanical nociception. *Drosophila* larvae display a stereotyped response to noxious mechanical insult. Importantly, this innate nocifensive behaviour, characterized by a 'corkscrew' body roll, can be quantified and is distinct from the animals' reaction to innocuous touch (*Figure 2A*; *Video 1*; *Tracey et al., 2003*; *Zhong et al., 2010*). Mechanical stimulation with a 40 mN von Frey filament triggered nocifensive behaviour in 53% of control larvae. In contrast, *Cirl* null mutants (*dCirl^{KO}*) showed significantly increased nocifensive behaviour and responded in 75% of the trials (*Figure 2B*; *Table 1*). Knocking-down *Cirl* levels specifically in C4da neurons via RNA-interference (RNAi; *ppk-GAL4 >UAS-dCirl^{RNAi}*) delivered a mutant phenocopy and re-expressing *Cirl* in nociceptors rescued the mutant phenotype. Notably, *Cirl* overexpression had the opposite effect resulting in diminished nocifensive responses (*Figure 2B*; *Table 1*). These results show that CIRL curtails mechanical nociception by carrying out a cell-autonomous, dose-dependent function in C4da neurons.

CIRL sensitizes proprioceptive ChO neurons by translating extracellular mechanical stimulation into a drop of intracellular cAMP. Lower cAMP levels, in turn, enhance the mechanically-evoked receptor potential of ChOs through a yet unresolved molecular mechanism (*Scholz et al., 2017*). Intriguingly, our behavioural data point towards CIRL exerting the opposite influence on nociceptive C4da neurons, whose sensitivity to mechanical stimulation is decreased by CIRL and increased in the absence of the aGPCR. A possible explanation for the antinociceptive action of CIRL is that the aGPCR also reduces cAMP in nociceptors, but that here the second messenger cascade acts on different molecular targets, i.e. specific ion channels, to produce an inverted effect. According to this

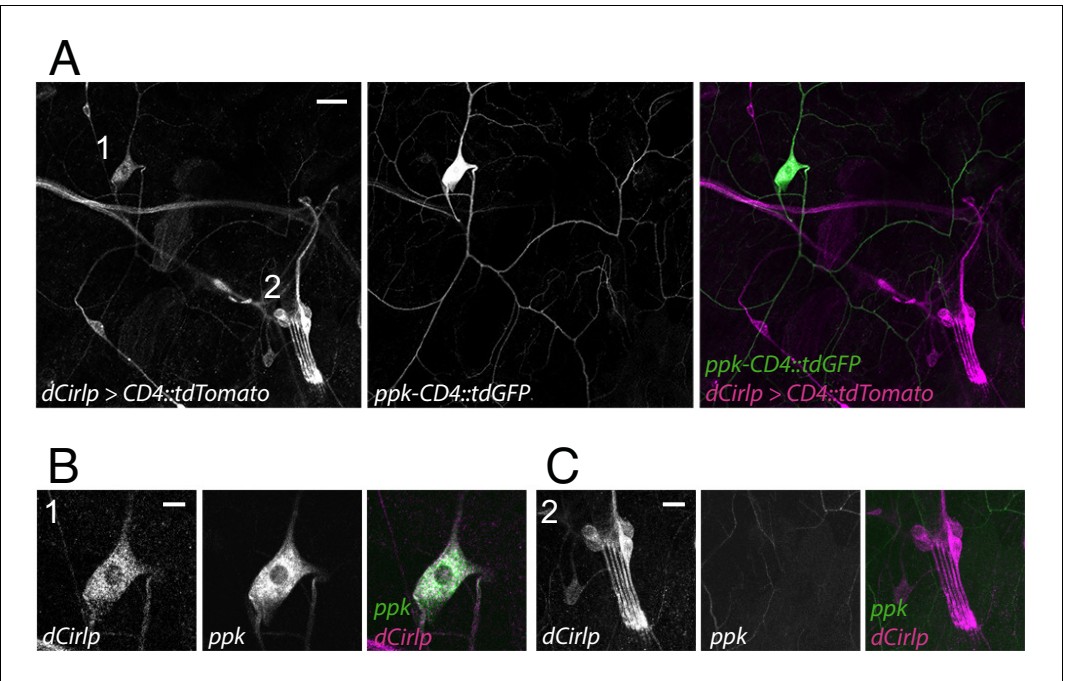

**Figure 1.** *Drosophila Cirl* is expressed in proprioceptors and nociceptors. (**A**) The *Cirl* promoter drives *Tomato* photoprotein expression (magenta; *dCirlp^{GAL4} >UAS-CD4::tdTomato*) in type one larval pentascolopidial ChO (lch5) neurons and type 2 C4da nociceptors, identified by a *GFP-ppk* promoter fusion (green; *ppk-CD4::tdGFP*). Magnified view of (**B**) C4da and (**C**) ChO neurons. Shown are immunohistochemical stainings against the fluorescent proteins. Scale bars (**A**) 20 μm, (**B,C**) 10 μm.

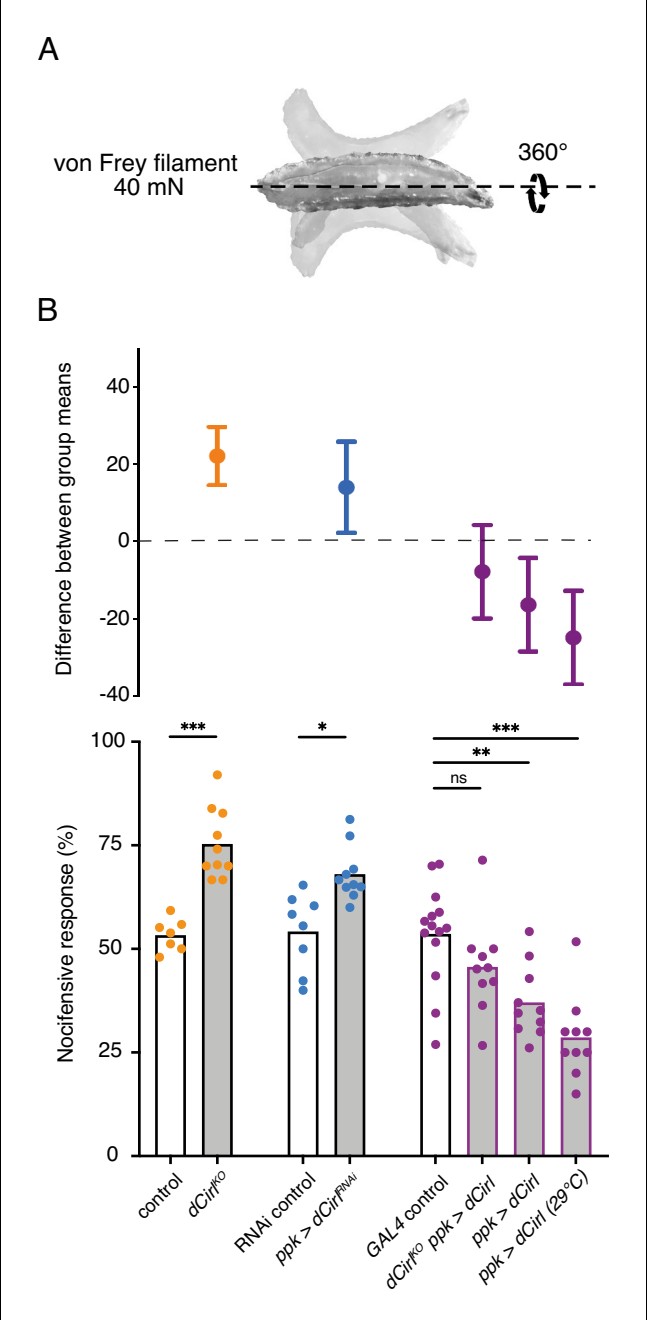

**Figure 2.** *Cirl* reduces nocifensive behaviour. (**A**) Characteristic nocifensive 'corkscrew' roll of larvae upon mechanical stimulation with a von Frey filament (40 mN force). (**B**) Quantification of nocifensive behaviour in different genotypes. Increased nocifensive responses were observed in *dCirl^{KO}* and upon nociceptor-specific expression of an RNAi construct (*ppk-GAL4 >UAS-dCirl^{RNAi}*). *Cirl* re-expression rescued the null mutant (*dCirl^{KO} ppk-GAL4 >UAS-dCirl*) and *Cirl* overexpression (*ppk-GAL4 >UAS-dCirl*) reduced nocifensive responses. Raising animals at a higher temperature (29°C vs. 25°C) increases *UAS/GAL4*-dependent transgene expression (***Duffy, 2002***). Data are presented as mean and individual values (lower bar plot) and as the difference between means with 95% confidence intervals (upper dot plot). Asterisks denote level of significance, *p≤0.05, **p≤0.01, ***p≤0.001.

model, low cAMP levels would dampen nociceptor activity. Next, we therefore asked whether increasing cAMP in C4da neurons (as may occur in *dCirl^{KO}*) promotes nocifensive behaviour. Because chronic changes of cAMP levels strongly alter neuronal development (***Griffith and Budnik, 2006***) we

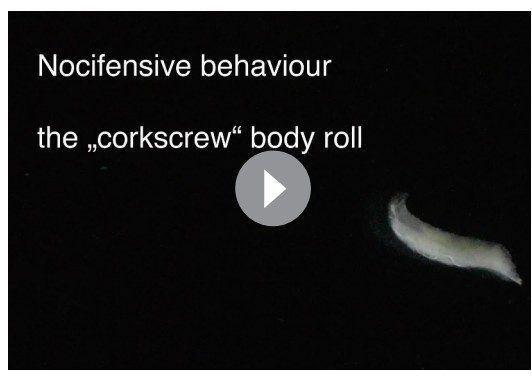

**Video 1.** Nocifensive behaviour - the 'corkscrew' body roll
https://elifesciences.org/articles/56738#video1

selected an optogenetic approach to trigger a rapid, nociceptor-specific increase of cAMP. The bacterial photoactivated adenylyl cyclase bPAC can be genetically expressed in selected *Drosophila* neurons to evoke cAMP production upon blue light stimulation (*Figure 3A*; *Stierl et al., 2011*; *Scholz et al., 2017*). Indeed, driving bPAC in C4da neurons (*ppk-GAL4 >UAS-bPAC*) led to increased nocifensive behaviour during light exposure, whereas control animals showed no light-induced effects (*Figure 3B*; *Table 2*). Notably, bPAC expression also produced some irregular behaviour in the absence of photostimulation ('spontaneous bending'; *Figure 3B*), which is likely a result of the enzyme's residual dark activity (*Stierl et al., 2011*). Nocifensive responses were further enhanced by bPAC in $dCirl^{KO}$ larvae ($dCirl^{KO}$ *ppk-GAL4 >UAS-bPAC*), in line with increased cAMP levels in mutant C4da neurons. Independent support for this conclusion was received by analysing larvae carrying mutant alleles of the phosphodiesterase (PDE) *dunce* (*Davis and Kiger, 1981*). Here, pronounced nocifensive behaviour accompanies chronically elevated cAMP concentrations (*Figure 3C*; *Table 1*).

To further substantiate that CIRL influences nociception by acting on cAMP-dependent signalling, we inhibited the endogenous adenylyl cyclase activity by pharmacological means. Consistent with nociceptor sensitization by cAMP, dietary supplementation with the adenylyl cyclase inhibitors SQ22536 or DDA (2',3'-dideoxyadenosine) significantly decreased nocifensive behaviour of *Drosophila* larvae. Importantly, this treatment produced the same responses of control and $dCirl^{KO}$ animals to von Frey filament stimulation (*Figure 3D*; *Table 1*). This result shows that CIRL acts in the same pathway as cAMP production by the adenylyl cyclase and supports the notion that the aGPCR exerts its antinociceptive effect by lowering cAMP levels through $G_{i/o}$-mediated inhibition of adenylyl cyclase activity.

Next, we used calcium imaging to directly test whether CIRL modulates the mechanically-evoked activity of nociceptors. To this end, we monitored calcium signals in C4da neurons labelled with GCaMP6m (*Chen et al., 2013*) during von Frey filament stimulation as previously reported (*Hu et al., 2017*; *Figure 4—figure supplement 1*). Consistent with the behavioural data describing a CIRL-mediated downregulation of nociceptor function, $dCirl^{RNAi}$ significantly enhanced calcium responses to noxious mechanical stimulation (*Figure 4A,B*; *Table 1*). In principle, CIRL could influence the activity of C4da neurons by modulating cellular excitability or by modulating the mechanotransduction process, as is the case in touch-sensitive ChO neurons (*Scholz et al., 2017*). To differentiate between these possibilities, we again chose an optogenetic strategy and circumvented mechanotransduction by stimulating the nociceptors with light. Photostimulation of C4da neurons via the optimized Channelrhodopsin-2 (ChR2) variant $ChR2^{XXM}$ (*Nagel et al., 2003*; *Scholz et al., 2017*) triggered nocifensive behaviour, consistent with previous work (*Hwang et al., 2007*). Notably, $dCirl^{KO}$ larvae ($dCirl^{KO}$ *ppk-GAL4 >UAS-chop2$^{XXM}$*) responded more strongly than controls (*ppk-GAL4 >UAS-chop2$^{XXM}$*) over a range of different blue light intensities (475 nm; *Figure 4C*; *Table 3*). This demonstrates that CIRL decreases the excitability of mechanical nociceptors, contrasting its role in touch-sensitive neurons where the aGPCR specifically enhances mechanotransduction (*Scholz et al., 2017*).

We further asked whether signalling to different subcellular targets may correlate with different activation mechanisms of CIRL in these two types of mechanosensory neurons. Several studies have shown that aGPCR activation can occur by means of an intramolecular tethered agonist, termed *Stachel* (stalk) (*Liebscher et al., 2014*; *Monk et al., 2015*; *Stoveken et al., 2015*, for *Stachel*-independent signalling see *Kishore et al., 2016*; *Salzman et al., 2017*; *Sando et al., 2019*). This linker sequence of approximately 20 amino acids begins at the extracellular GPCR proteolysis site (GPS) and connects the GAIN (GPCR autoproteolysis inducing) domain to the 7-transmembrane unit (*Figure 4D*). In order to test for a role of the *Stachel* sequence in activating CIRL in mechanical nociceptors, we used two established mutants. Whereas the $dCirl^{T>A}$ allele carries a mutation at the +1

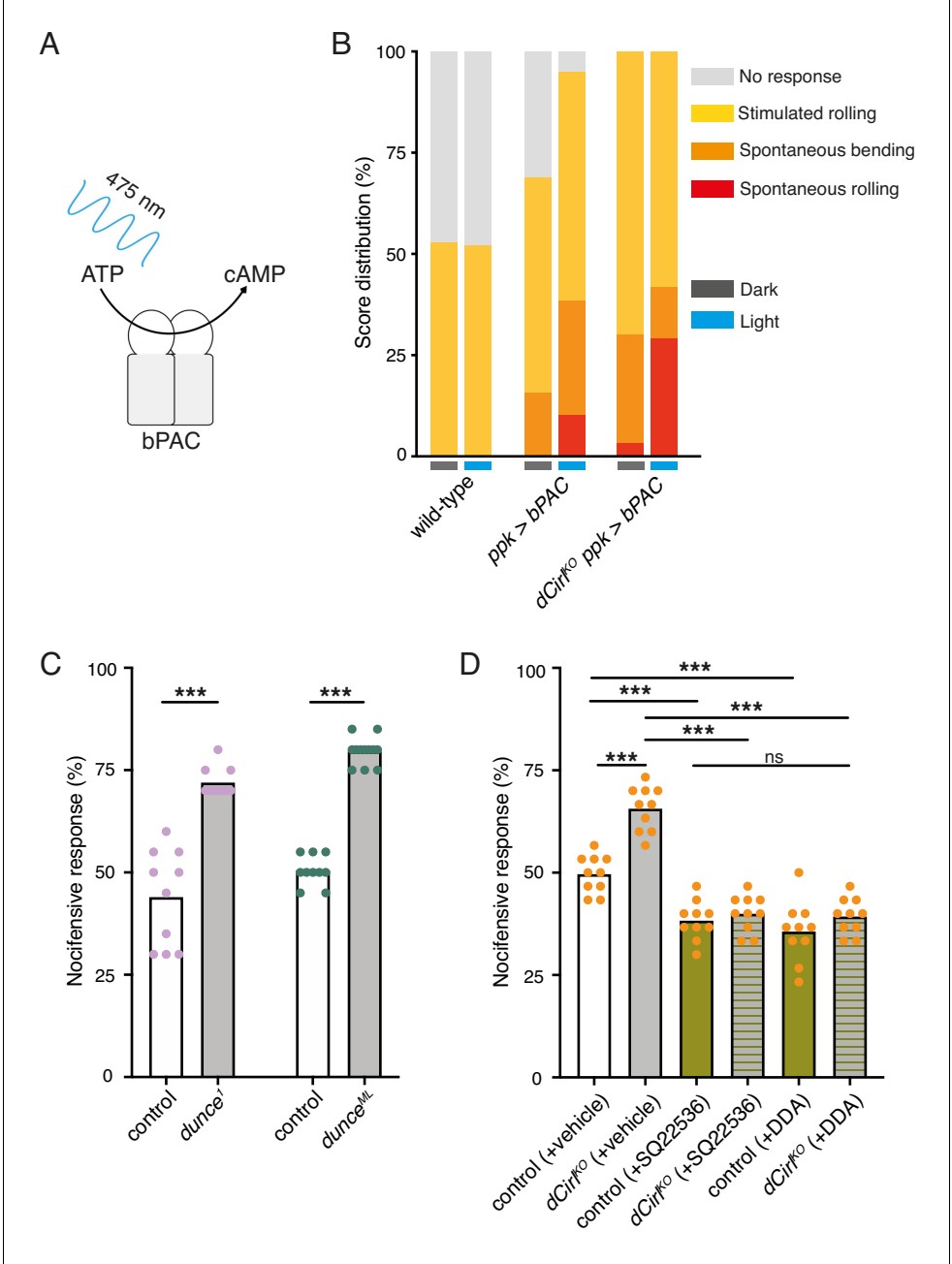

**Figure 3.** Potentiation of nociceptor function by cAMP. (**A**) Schematic illustration of cAMP production by bPAC. (**B**) Optogenetic assay. Stimulated and spontaneous nocifensive responses can be promoted and elicited, respectively, by bPAC activation in C4da neurons (blue labels, photostimulation). Larval behaviour was observed during 3 min illumination (~200 μW/mm$^2$ at 475 nm) followed by mechanical stimulation (40 mN von Frey filament). (**C**) Nocifensive behaviour of PDE mutants with ~73% (*dunce*$^1$) and ~35% (*dunce*$^{ML}$) residual cAMP hydrolysis rates (*Davis and Kiger, 1981*). (**D**) The adenylyl cyclase inhibitors SQ22536 and DDA (500 μM) reduce nocifensive responses to comparable levels in control and *dCirl*$^{KO}$ larvae. Data are presented as mean and individual values. Asterisks denote level of significance, ***p≤0.001.

position of the GPS within the *Stachel* sequence, the −2 mutation in the *dCirl*$^{H>A}$ allele leaves the *Stachel* intact but reduces protein expression (*Figure 4D*; *Scholz et al., 2017*). Our analysis of nocifensive behaviour in these mutants reported normal responses for the T > A allele and a *Cirl*$^{KO}$ phenocopy for the H > A allele (*Figure 4E*, *Table 1*). Thus, C4da neurons appear to be sensitive to

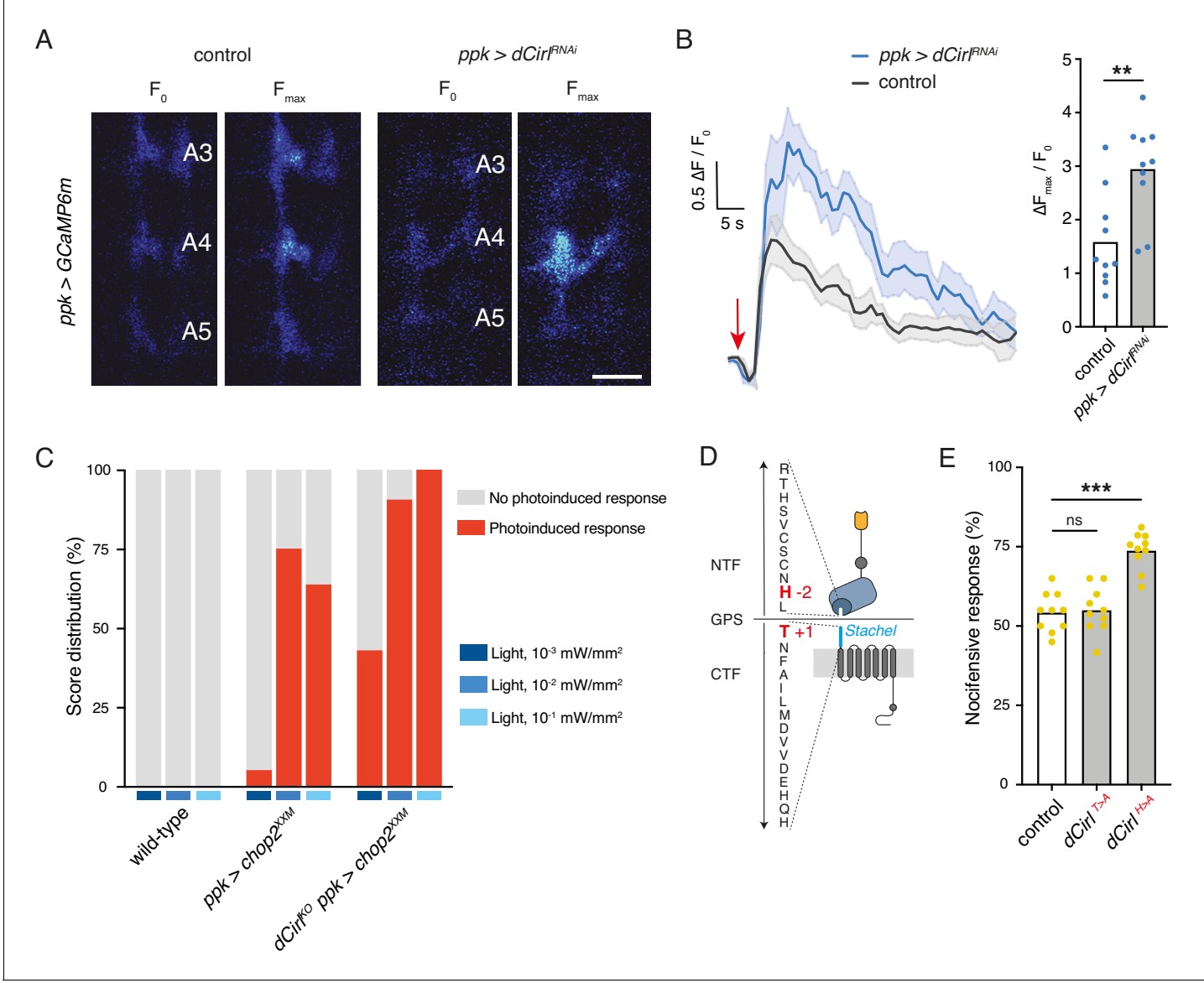

**Figure 4.** *Cirl* decreases the excitability of nociceptors. (**A**) Calcium imaging of C4da axon terminals expressing GCaMP6m (*ppk-GAL4 >UAS-GCaMP6m*) in semi-intact larval preparations. Representative baseline ($F_0$) and maximum calcium responses ($F_{max}$) are shown for control and $Cirl^{RNAi}$ animals upon von Frey filament stimulation (45 mN). Scale bar, 10 μm. (**B**) Average calcium traces (arrow indicates stimulation) and quantification of the signals ($\Delta F_{max}/F_0$). $Cirl^{RNAi}$ significantly elevates mechano-nociceptive responses of C4da neurons. (**C**) Nocifensive responses (red) elicited via $ChR2^{XXM}$-mediated photostimulation of C4da neurons in control (*ppk-GAL4 >UAS-chop2^{XXM}*) and $dCirl^{KO}$ larvae ($dCirl^{KO}$ *ppk-GAL4 >UAS-chop2^{XXM}*). (**D**) Structure of the GPS region in *Drosophila* CIRL (*Scholz et al., 2017*). The *Stachel* sequence (light blue) is part of the GAIN domain (blue) contained in the CTF. Conserved, mutated amino acids required for receptor autoproteolysis at the GPS are shown in red (−2: $dCirl^{H>A}$, +1: $dCirl^{T>A}$). (**E**) Quantification of nocifensive behaviour in $dCirl^{T>A}$ and $dCirl^{H>A}$ receptor mutants. Data are presented as mean and individual values. Asterisks denote level of significance, **$p \leq 0.01$, ***$p \leq 0.001$. See also *Figure 4—figure supplements 1* and *2*.

The online version of this article includes the following figure supplement(s) for figure 4:

**Figure supplement 1.** Larval preparation for calcium imaging.

**Figure supplement 2.** CIRL protein expression in mechanical nociceptors.

reduced CIRL expression, which is consistent with very low protein levels in wild-type animals (close to the detection threshold; *Figure 4—figure supplement 2*). An intact tethered agonist, in turn, is dispensable for CIRL function in mechanical nociceptors, contrasting the situation in touch-sensitive ChO neurons, where mutating the *Stachel* sequence disrupts receptor function (*Scholz et al., 2017*).

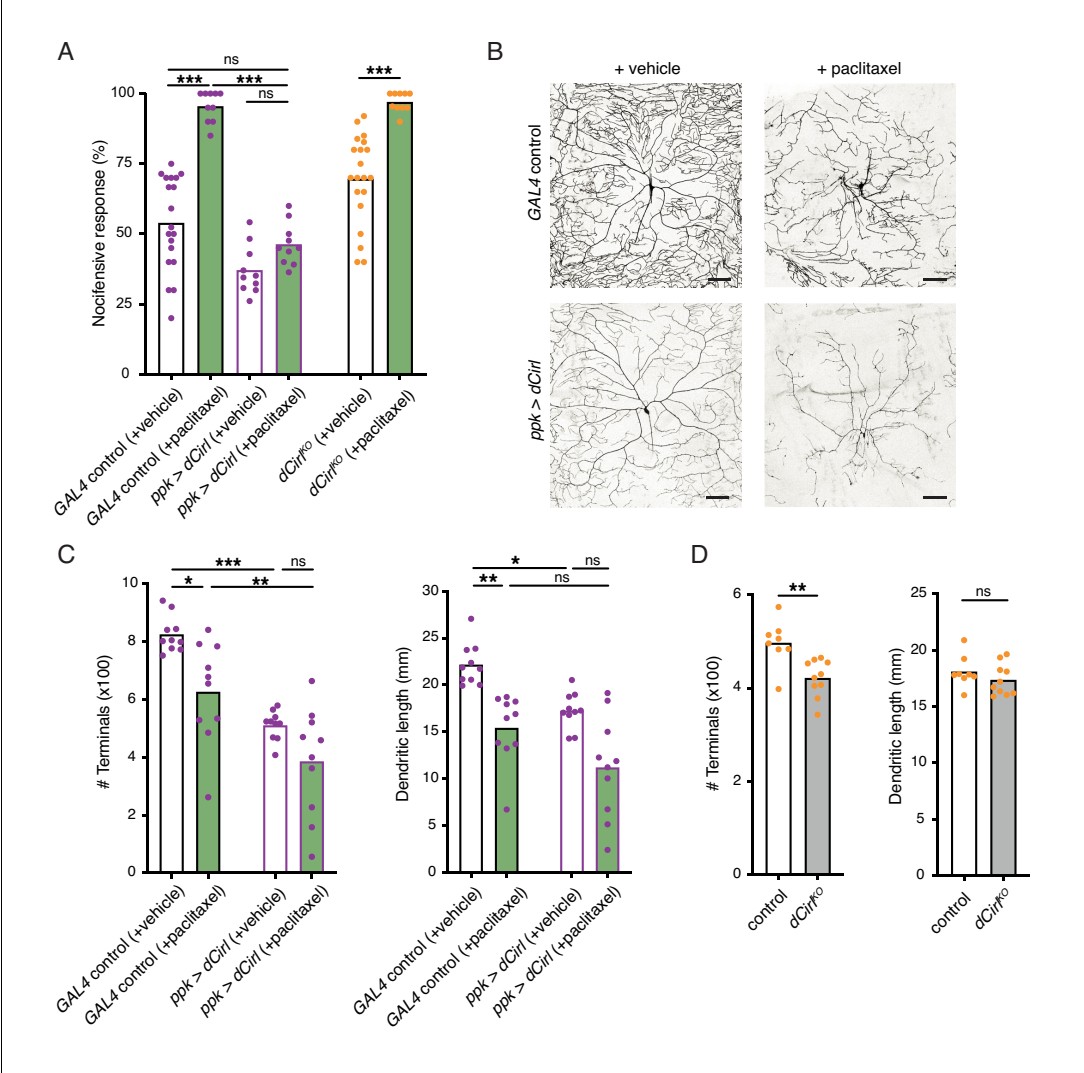

**Figure 5.** Sensitization of nociceptors through chemotherapy-induced neuropathy. (**A**) Increased nocifensive behaviour following paclitaxel treatment (10 µM) is counteracted by overexpressing *Cirl* in nociceptors. (**B**) Example images of C4da neuron morphology upon paclitaxel administration and *Cirl* overexpression. Scale bars, 100 µm. (**C, D**) Morphometric quantification of dendritic complexity of C4da neurons in the different genotypes. Data are presented as mean and individual values. Asterisks denote level of significance, *p≤0.05, **p≤0.01, ***p≤0.001.

Taken together, these observations indicate that the activation mechanism of CIRL differs for the two mechanosensory submodalities.

Having established that CIRL downregulates nociceptor function under physiological conditions, we sought to investigate a pathological setting. The chemotherapeutic agent paclitaxel, employed to treat solid tumours such as ovarian or breast cancer, causes dose-limiting peripheral neuropathy in patients. Similarly, feeding *Drosophila* larvae paclitaxel-supplemented food induces axonal injury and degeneration of C4da neurons (*Bhattacharya et al., 2012*). Next, we therefore examined nocifensive behaviour in the context of this established neuropathy model. Consistent with chemotherapy-induced allodynia in humans, paclitaxel strongly enhanced nocifensive responses of control larvae (*Figure 5A*). We observed a comparable effect in *dCirl^{KO}* animals. Overexpressing *Cirl*, in turn, reverted the paclitaxel-induced sensitization of C4da neurons (*Figure 5A*; *Table 1*). Thus, CIRL tones down nociceptors in both physiological and neuropathic hyperexcitable states.

Paclitaxel administration causes structural damage to C4da neurons (*Bhattacharya et al., 2012*). Following paclitaxel treatment (10 µM), we observed dendrite loss in the wild-type background and in nociceptors overexpressing *Cirl* (*Figure 5B,C*; *Table 1*). In fact, elevated *Cirl* expression itself

reduced the dendritic complexity of C4da neurons. Thus, increasing CIRL protein copy number counteracts the neuropathic hyperexcitability of mechanical nociceptors independently of neuropathy-associated morphological defects. Consistent with the interpretation that modulation of nociceptor physiology by CIRL is not tightly coupled to morphological changes, *dCirl^KO* C4da neurons displayed only subtle structural alterations (*Figure 5D*).

Considering the evolutionary conservation of signalling pathways for nociception (*Im and Galko, 2012*), we next turned to a rodent model of traumatic neuropathic pain: unilateral chronic constriction injury (CCI) of the sciatic nerve (*Reinhold et al., 2019*). This model resembles paclitaxel-induced neuropathy in the development of thermal hypersensitivity and mechanical allodynia (*Sisignano et al., 2016*), i.e. a noxious reaction to innocuous stimuli like touch, reaching a maximum after one week (*Figure 6A*; *Table 1*). There are three CIRL proteins in mammals (CIRL1-3 also known as ADGRL1-3 or Lphn1-3) (*Langenhan et al., 2016*). According to RNA sequencing data from mouse DRG neurons, *Cirl1* and *Cirl3* are expressed in nociceptors (*Thakur et al., 2014*). We therefore investigated *Cirl1* and *Cirl3* expression in subpopulations of DRG neurons via in situ hybridization in the neuropathic context. Interestingly, *Cirl1* mRNA probes described significantly reduced transcript levels in isolectin-B4 (IB4)-positive, non-peptidergic nociceptors one week after CCI (*Figure 6B,D,E*; *Table 1*). We observed a similar, though statistically insignificant trend in peptidergic nociceptors identified by calcitonin gene-related peptide (CGRP) staining and for *Cirl3* probes (Figure 6B-E; *Table 1*). Notably, CCI appeared to neither affect *Cirl1* nor *Cirl3* gene expression in non-nociceptive, large myelinated neurons marked by neurofilament protein NF200. These correlative results are consistent with the *Drosophila* data linking low *Cirl* expression levels to nociceptor sensitization. It remains to be determined whether expression of *Drosophila Cirl* can be regulated physiologically to adapt or tune nociceptor sensitivity and whether activation of mammalian CIRL, in turn, can provide analgesia.

## Discussion

The sensations of touch and mechanical pain represent distinct mechanosensory submodalities, which are separated at the initial sites of mechanotransduction. Despite their important roles in health and disease, an understanding of how these mechanistically different transduction processes are carried out at the molecular level is only just beginning to emerge (*Delmas et al., 2011*; *Julius, 2013*; *Murthy et al., 2017*; *Szczot et al., 2017*; *Zhang et al., 2019*). Similar to mammalian DRG neurons, *Drosophila* nociceptors are modulated by GPCRs (*Hu et al., 2017*; *Kaneko et al., 2017*; *Herman et al., 2018*) and can be negatively regulated by $G_{i/o}$ signalling (*Christianson et al., 2016*; *Honjo et al., 2016*). In the present study, we provide evidence that CIRL, an evolutionarily conserved aGPCR, reduces nociceptor responses to mechanical insult in *Drosophila* larvae. This modulation operates in the opposite direction to the sensitization of touch sensitive neurons by CIRL (*Figure 7*; *Scholz et al., 2015*; *Scholz et al., 2017*). In both types of mechanosensors, these effects are connected to CIRL-dependent decreases of cAMP levels. The opposing cell physiological outcomes, in turn, likely arise from specific adjustments of different effector proteins through cAMP-signalling. Candidate effectors are mechanotransduction channels and ion channels, which are mechanically-insensitive but influence the rheobase, i.e. the threshold current of the sensory neuron (*Boiko et al., 2017*).

The transient receptor potential (TRP) channel subunits NOMPC (*no receptor potential,* TRPN), NAN (*nanchung*, TRPV), and IAV (*inactive*, TRPV) govern mechanosensation by larval ChO neurons (*Effertz et al., 2012*; *Lehnert et al., 2013*; *Zhang et al., 2013*). The mechanically gated ion channel Piezo, the DEG/ENaC subunit Pickpocket, and the TRPN channel Painless, on the other hand, are required for mechanical nociception in *Drosophila* (*Tracey et al., 2003*; *Zhong et al., 2010*; *Kim et al., 2012*; *Gorczyca et al., 2014*; *Guo et al., 2014*; *Mauthner et al., 2014*). It is therefore conceivable that the receptor potential generated by these different mechanotransducers may be modulated in opposite directions, i.e. decreased in ChO neurons and increased in nociceptors, by cAMP/PKA (protein kinase A)-dependent channel phosphorylation. Matching our results in *Drosophila*, enhanced nociceptor activity in mammals has been linked to elevated cAMP levels. For example, mechanical hyperalgesia during inflammation involves cAMP-modulated HCN channels and sensitization of mammalian Piezo2 via PKA and protein kinase C (PKC)-based signalling (*Emery et al., 2011*; *Dubin et al., 2012*). Conversely, $G_{i/o}$-coupled receptors, such as opioid, somatostatin, and GABA$_B$

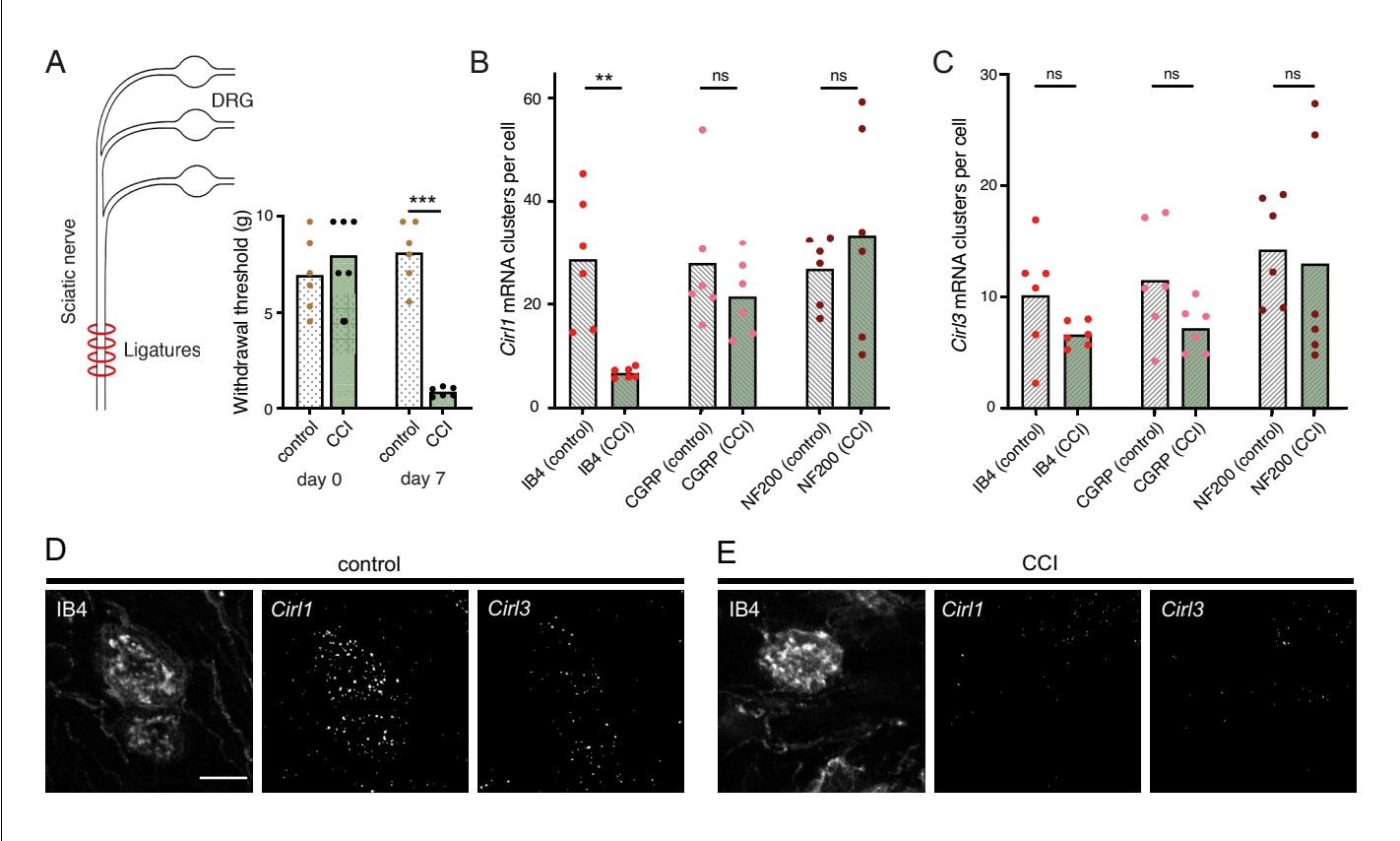

**Figure 6.** Neuropathy-induced mechanical allodynia correlates with decreased *Cirl1* expression in mammalian non-peptidergic nociceptors. (A) Traumatic injury of the sciatic nerve (CCI, green) in Wistar rats results in mechanical allodynia after one week as measured by von Frey Hairs (paw withdrawal threshold) in comparison to the contralateral side (grey). (B, C) Quantification of *Cirl1* (B) and *Cirl3* (C) mRNA levels in subpopulations of rat DRG neurons via in situ hybridization (RNAscope). Shown are control conditions (naïve DRGs, grey) and one week after injury (green) following the emergence of allodynia. Data are presented as mean and individual values. Asterisks denote level of significance, **p≤0.01, ***p≤0.001. (D, E) Example images of the RNAscope assay in DRG neurons. Shown are projections of confocal stacks stained against IB4 and labelled with probes against *Cirl1* and *Cirl3* under control conditions (D) and CCI (E). Scale bar 20 μm.

receptors, counteract cAMP-dependent nociceptor sensitization (*Yudin and Rohacs, 2018*). In addition to this second messenger pathway, $G_{\beta\gamma}$ subunits of $G_{i/o}$-coupled GPCRs can directly interact with ion channels. Thereby nociceptor signalling can be suppressed via activation of G protein regulated inwardly rectifying K$^+$ channels (GIRK) or by inhibition of voltage-gated Ca$^{2+}$ channels (*Logothetis et al., 1987*; *Marker et al., 2005*; *Bourinet et al., 2014*). Recent work has identified that CIRL2 and CIRL3 promote synapse formation in the mouse hippocampus (*Sando et al., 2019*). While *Drosophila* CIRL may also shape synaptic connectivity, our results indicate that CIRL modulates the mechanically-evoked activity of nociceptors independently of such an additional function.

The present findings show that CIRL decreases the activity of C4da neurons independently of mechanotransduction and that the aGPCR feeds into the same pathway as the adenylyl cyclase. Taken together, this strongly suggests that $G_{i/o}$ coupling by CIRL regulates cAMP-dependent modulation of ion channels, which control nociceptor excitability. Work in cell culture has put forward a model in which *Stachel*-dependent and -independent aGPCR activation triggers different downstream signalling pathways (*Kishore et al., 2016*; *Salzman et al., 2017*). Here, we provide evidence in support of such a dual activation model in a physiological setting. The dispensability of an intact *Stachel* sequence in mechanical nociceptors and its necessity in touch-sensitive neurons argues for alternative activation modes of CIRL in these two types of mechanosensory neurons. This raises the intriguing possibility that such functional differentiation may be connected to specific downstream effects, e.g. *Stachel*-dependent, phasic modulation of mechanotransduction in ChO neurons versus *Stachel*-independent, tonic modulation of nociceptor excitability (*Figure 7*).

**Table 1.** Behaviour and imaging.

| Figure | Genotype | Mean | SEM | N | P-value | | Test |
|---|---|---|---|---|---|---|---|
| *Figure 2B* | control | 53.34 | 1.458 | 7 | p<0.0001 | | unpaired t-test |
| | $dCirl^{KO}$ | 75.37 | 2.676 | 10 | | | |
| | RNAi control | 54.23 | 3.279 | 8 | p=0.0169 | | one-way ANOVA, Tukey correction |
| | $ppk > dCirl^{RNAi}$ | 68.08 | 2.052 | 10 | | | |
| | *GAL4* control | 53.67 | 3.211 | 14 | | | one-way ANOVA, Tukey correction |
| | KO $ppk > dCirl$ | 45.7 | 3.627 | 10 | p=0.3345 (*GAL4* control) | | |
| | $ppk > dCirl$ | 37.12 | 2.778 | 10 | p=0.0023 (*GAL4* control) | | |
| | $ppk > dCirl$ (29°C) | 28.67 | 3.125 | 10 | p<0.0001 (*GAL4* control) | | |
| *Figure 3C* | control (Canton-S) | 44 | 3.712 | 10 | p<0.0001 | | Mann-Whitney |
| | $dunce^1$ | 72 | 1.106 | 10 | | | |
| | control ($f^{36a}$) | 50.5 | 1.167 | 10 | p<0.0001 | | unpaired t-test |
| | $dunce^{ML}$ | 79.58 | 0.965 | 12 | | | |
| *Figure 3D* | control (+vehicle) | 49.67 | 1.445 | 10 | p<0.0001 | | one-way ANOVA, Tukey correction |
| | $dCirl^{KO}$(+vehicle) | 65.67 | 1.725 | 10 | | | |
| | control (+SQ22536) | 38.33 | 1.511 | 10 | p=0.9805 | p=0.0002 (control +vehicle) | |
| | $dCirl^{KO}$(+SQ22536) | 40 | 1.406 | 10 | | p<0.0001 ($dCirl^{KO}$ +vehicle) | |
| | control (+DDA) | 35.67 | 2.334 | 10 | p=0.6315 | p<0.0001 (control +vehicle) | |
| | $dCirl^{KO}$(+DDA) | 39.33 | 1.388 | 10 | | p<0.0001 ($dCirl^{KO}$ +vehicle) | |
| *Figure 4B* | control | 1.58 | 0.2794 | 10 | p=0.0032 | | unpaired t-test |
| | $ppk > dCirl^{RNAi}$ | 2.94 | 0.2866 | 10 | | | |
| *Figure 4E* | control | 54.28 | 1.961 | 10 | | | one-way ANOVA, Tukey correction |
| | $dCirlT^{T>A}$ | 55 | 2.274 | 10 | p=0.9660 (control) | | |
| | $dCirlH^{H>A}$ | 73.78 | 1.845 | 10 | p<0.0001 (control) | | |
| *Figure 5A* | *GAL4* control (+vehicle) | 53.96 | 3.831 | 19 | p>0.9999 ($ppk > dCirl$ +paclitaxel) | | Kruskal-Wallis test |
| | *GAL4* control (+paclitaxel) | 95.5 | 1.74 | 10 | p=0.0002 (*GAL4* control +vehicle) | | |
| | $ppk > dCirl$ (+vehicle) | 37.12 | 2.778 | 10 | p>0.9999 ($ppk > dCirl$ +paclitaxel) | | |
| | $ppk > dCirl$ (+paclitaxel) | 46.34 | 2.374 | 10 | p<0.0001 (*GAL4* control +Taxol) | | |
| | $dCirl^{KO}$ (+vehicle) | 69.68 | 3.54 | 20 | p<0.0001 | | Mann-Whitney |
| | $dCirl^{KO}$ (+paclitaxel) | 97.05 | 1.097 | 10 | | | |
| *Figure 5C* | *GAL4* control (+vehicle) | 8.256 | 0.1985 | 10 | p=0.0117 (*GAL4* control +Taxol) | | Kruskal-Wallis test |
| | *GAL4* control (+paclitaxel) | 6.27 | 0.5574 | 10 | p=0.0017 ($ppk > dCirl$ +paclitaxel) | | |
| | $ppk > dCirl$ (+vehicle) | 5.108 | 0.1603 | 10 | p<0.0001 (*GAL4* control +vehicle) | | |
| | $ppk > dCirl$ (+paclitaxel) | 3.861 | 0.5961 | 10 | p=0.1848 ($ppk > dCirl$ +vehicle) | | |
| | *GAL4* control (+vehicle) | 22.17 | 0.7015 | 10 | p=0.0014 (*GAL4* control +paclitaxel) | | one-way ANOVA, Tukey correction |
| | *GAL4* control (+paclitaxel) | 15.42 | 1.174 | 10 | p>0.9999 (ppk > dCirl +paclitaxel) | | |
| | $ppk > dCirl$ (+vehicle) | 17.26 | 0.6091 | 10 | p=0.0296 (*GAL4* control +vehicle) | | |
| | $ppk > dCirl$ (+paclitaxel) | 11.22 | 1.716 | 10 | p=0.2557 ($ppk > dCirl$ +vehicle) | | |
| *Figure 5D* | control | 4.971 | 0.1747 | 8 | p=0.0025 | | unpaired t-test |
| | $dCirl^{KO}$ | 4.219 | 0.1255 | 10 | | | |
| | control | 18.11 | 0.5029 | 8 | p=0.2829 | | unpaired t-test |
| | $dCirl^{KO}$ | 17.37 | 0.4384 | 10 | | | |

*Table 1 continued on next page*

*Table 1 continued*

| Figure | Genotype | Mean | SEM | N | P-value | Test |
|---|---|---|---|---|---|---|
| *Figure 6A* | control day 0 | 6.972 | 0.8056 | 6 | p=0.4062 | unpaired t-test |
| | CCI day 0 | 8.003 | 0.8755 | 6 | | |
| | control day 7 | 8.152 | 0.6647 | 6 | p<0.0001 | |
| | CCI day 7 | 0.861 | 0.0982 | 6 | | |
| *Figure 6B* | IB4 (control) | 28.6 | 5.148 | 6 | p=0.0018 | unpaired t-test |
| | IB4 (CCI) | 6.791 | 0.4034 | 6 | | |
| | CGRP (control) | 27.89 | 5.529 | 6 | p=0.3358 | |
| | CGRP (CCI) | 21.49 | 3.078 | 6 | | |
| | NF200 (control) | 26.76 | 2.737 | 6 | p=0.45 | |
| | NF200 (CCI) | 33.57 | 8.216 | 6 | | |
| *Figure 6C* | IB4 (control) | 10.16 | 2.065 | 6 | p=0.1293 | unpaired t-test |
| | IB4 (CCI) | 6.662 | 0.4542 | 6 | | |
| | CGRP (control) | 11.5 | 2.103 | 6 | p=0.0895 | |
| | CGRP (CCI) | 7.212 | 0.8851 | 6 | | |
| | NF200 (control) | 14.25 | 1.965 | 6 | p=0.3939 | Mann-Whitney |
| | NF200 (CCI) | 13 | 4.133 | 6 | | |

Many genes display altered expression in DRG neurons in neuropathy (*Lopes et al., 2017*). For example, receptors and ion channels involved in sensitization are upregulated, whereas endogenous antinociceptive mechanisms, including opioid receptors and their peptides, are downregulated in certain neuropathy models (*Herradon et al., 2008*; *Hervera et al., 2011*). Thus, neuropathy not only enhances pro-nociceptive mechanisms but also decreases endogenous antinociceptive pathways. Our analysis of rodent DRGs indicates that neuropathy-induced allodynia correlates with reduced *Cirl1* expression in IB4-positive non-peptidergic nociceptors (*Fang et al., 2006*), a class of neurons, which have been linked to mechanical inflammatory hypersensitivity (*Pinto et al., 2019*). It is therefore tempting to speculate that CIRL operates via a conserved antinociceptive mechanism in both invertebrate and mammalian nociceptors to reduce cAMP concentrations. Future work will have to test this hypothesis by examining a direct causal relation between CIRL activation and nociceptor attenuation in the mammalian peripheral nervous system and to explore whether metabotropic mechanosensing by CIRL is a possible target for analgesic therapy. Limited options for treating chronic pain have contributed to the current opioid epidemic (*Skolnick and Volkow, 2016*). Opioids are powerful analgesics but have severe side effects and lead to addiction mainly through activation of receptors in the central nervous system. There is thus a strong incentive to develop novel peripherally acting pain therapeutics.

The specificity theory, put forward more than 100 years ago (*Sherrington, 1906*), defines nociceptors as a functionally distinct subtype of nerve endings, which are specifically tuned to detect harmful, high-intensity stimuli. The results reported in the present study are consistent with this validated concept and identify a physiological mechanism, which contributes to the functional specialization. On the one hand, CIRL helps set the high activation threshold of mechanical nociceptors, while on the other hand, CIRL lowers the activation threshold of touch sensitive neurons (*Figure 7*). This bidirectional adjustment, attributable to cell-specific effects of cAMP, moves both submodalities further apart and sharpens the contrast of mechanosensory signals carrying different information.

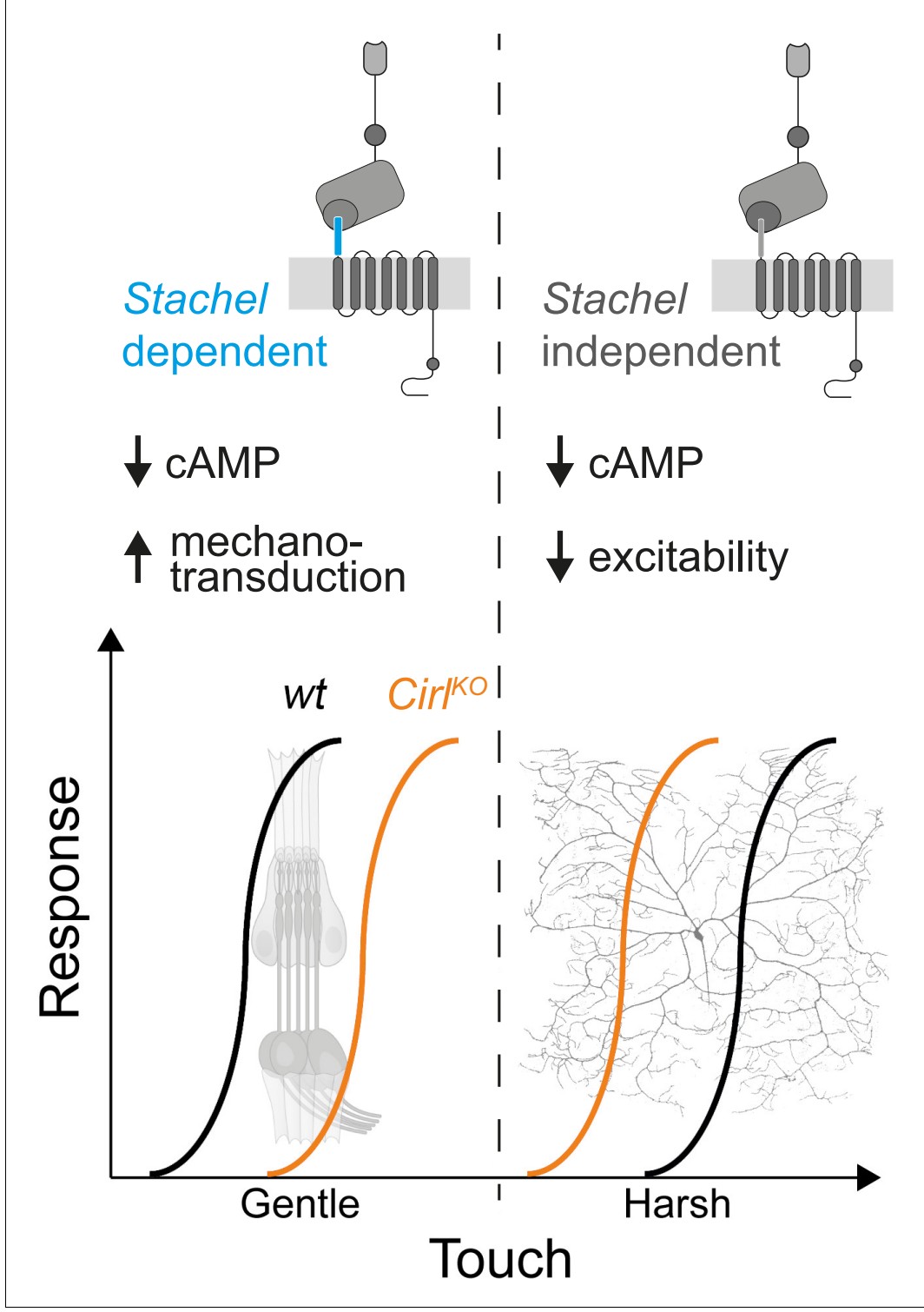

**Figure 7.** cAMP downregulation by *Drosophila* CIRL adjusts mechanosensory submodalities in opposite directions. Scheme summarizing how processing of different levels of mechanical force is bidirectionally modulated by CIRL's downregulation of cAMP production. Whereas low threshold mechanosensory neurons (ChOs; gentle touch) are less responsive in *Cirl* mutants, high threshold mechanical nociceptors (C4da neurons; harsh touch) become sensitized.

## Materials and methods

### *Drosophila* experiments

Fly stocks

Animals were raised at 25°C on standard cornmeal and molasses medium. The following fly strains were used in this study:

UAS-dCirl$^{RNAi}$ (VDRC#100749): $w^{1118}$; phiC31{KK108383}v100749 (*Dietzl et al., 2007*)
$w^{1118}$; UAS-bpac/CyOGFP $w^-$ (*Stierl et al., 2011*)

from *Scholz et al., 2015*; *Scholz et al., 2017*:
dCirlp$^{GAL4}$ (LAT84): $w^{1118}$; dCirl$^{KO}$ {$w^{+mC}$ = pTL464[dCirlp::GAL4]}attP$^{dCirl}$ loxP/CyOGFP $w^-$
dCirl$^{KO}$ (LAT26): $w^{1118}$; dCirl$^{KO}$ attP$^{dCirl}$ loxP
dCirl$^{Rescue}$ (LAT163): $w^{1118}$; dCirl$^{KO}$ {$w^{+mC}$ = pTL370[dCirl]}attP$^{dCirl}$ loxP
20xUAS-dCirl (LAT85): $w^{1118}$;; {$w^{+mC}$ = pTL471[20xUAS-IVS-dCirl::3xFlag]}attP2
dCirl$^{KO}$ 20xUAS-dCirl (LAT111): $w^{1118}$; dCirl$^{KO}$ attP$^{dCirl}$ loxP; {$w^{+mC}$ = pTL471[20xUAS-IVS-dCirl::3xFlag]}attP2/TM6B, Tb
dCirl$^{T>A}$ (LAT174): $w^{1118}$; dCirl$^{KO}$ {$w^{+mC}$ = pMN9[dCirl$^{T>A}$]}attP$^{dCirl}$loxP/CyoGFP $w^-$
dCirl$^{H>A}$ (LAT280): $w^{1118}$; dCirl$^{KO}$ {$w^{+mC}$ = pMN44[dCirl$^{H>A}$]}attP$^{dCirl}$loxP/CyoGFP $w^-$
dCirl$^{N-RFP}$ (LAT159): $w^{1118}$; dCirl$^{KO}$ {$w^{+mC}$ = pMN4[dCirl$^{N-RFP}$]}attP$^{dCirl}$loxP/CyoGFP $w^-$
UAS-chop2$^{XXM}$ (RJK300): $w^{1118}$; {$w^{+mC}$ = pTL537[chop2-D156H(XXM)::tdtomato]}attP$^{VK00018}$/CyoGFP $w^-$
dCirl$^{KO}$ UAS-chop2$^{XXM}$ (LAT193): $w^{1118}$; dCirl$^{KO}$ {$w^{+mC}$ = pTL537[chop2-D156H(XXM)::tdtomato]}attP$^{VK00018}$/CyoGFP $w^-$

from the Bloomington *Drosophila* Stock Center:
BDSC#42748: $w^{1118}$; UAS-GCaMP6m P{$y^{+t7.7}w^{+mC}$ = 20xUAS-IVS-GCaMP6m}attP40
BDSC#35843: $w^{1118}$;; P{$w^{+mC}$ = ppk-CD4::tdGFP}8/TM6B,Tb
BDSC#35841: $y^1$ $w^*$; P{$w^{+mC}$ = UAS-CD4::tdTomato}7 M1
BDSC#32078: $w^*$; P{$w^{+mC}$ = ppk-GAL4.G}2
BDSC#32079: $w^*$;; P{$w^{+mC}$ = ppk-GAL4.G}3
BDSC#6020: dnc$^1$
BDSC#43: $f^{36a}$
BDSC#9407: $y^1$ $w^1$ dnc$^{ML}$ $f^{36a}$/FM7a

*Figure 1*:
dCirlp$^{GAL4}$/UAS-CD4::tdTomato; ppk-CD4::tdGFP/+

*Figure 2*:
control: dCirl$^{Rescue}$
dCirl$^{KO}$
RNAi control: UAS-dCirl$^{RNAi}$/+
ppk >dCirl$^{RNAi}$: ppk-GAL4/UAS-dCirl$^{RNAi}$
GAL4 control: dCirl$^{Rescue}$/+; ppk-GAL4/+
dCirl$^{KO}$ ppk >dCirl: dCirl$^{KO}$; ppk-GAL4/20xUAS-dCirl
ppk >dCirl: ppk-GAL4/20xUAS-dCirl

*Figure 3*:
(B) wild-type: $w^{1118}$
ppk >bPAC: ppk-GAL4/UAS-bpac
dCirl$^{KO}$ ppk >bPAC: dCirl$^{KO}$, UAS-bpac/dCirl$^{KO}$; ppk-GAL4/+
(C) control for *dunce$^1$*: Canton-S
control for *dunce$^{ML}$*: $f^{36a}$
(D) control: dCirl$^{Rescue}$
dCirl$^{KO}$

*Figure 4*:
(A,B) control: UAS-GCaMP6m/+; ppk-GAL4/+

ppk >dCirl$^{RNAi}$: UAS-GCaMP6m/UAS-dCirl$^{RNAi}$; ppk-GAL4/+
(C) wild-type: w$^{1118}$
ppk >chop2$^{XXM}$: UAS-chop2$^{XXM}$/+; ppk-GAL4/+
dCirl$^{KO}$ ppk >chop2$^{XXM}$: dCirl$^{KO}$ UAS-chop2$^{XXM}$/dCirl$^{KO}$; ppk-GAL4/+
(E) control: dCirl$^{Rescue}$
dCirl$^{T>A}$
dCirl$^{H>A}$

*Figure 5*:
 (A) *GAL4 control: ppk-GAL4/+*
 *ppk >dCirl: ppk-GAL4/20xUAS-dCirl*
 *dCirl$^{KO}$*
 (B, C) *GAL4 control: ppk-GAL4/+; ppk-CD4::tdGFP/+*
 *ppk >dCirl: ppk-GAL4/+; ppk-CD4::tdGFP/20xUAS-dCirl*
 (D) *Control: ppk-Gal4, UAS-CD4::tdTomato/+*
 *dCirl$^{KO}$: dCirl$^{KO}$; ppk-Gal4, UAS-CD4::tdTomato/+*

## Immunohistochemistry

Stainings of the *dCirlp$^{GAL4}$*- and *ppk*-positive neurons (*Figure 1*) and CIRL$^{N-RFP}$ (*Figure 4—figure supplement 2*) were performed essentially as previously reported (*Ehmann et al., 2014*). Third instar larvae were dissected in cold PBS, fixed in 4% paraformaldehyde for 10 or 40 min at room temperature (RT), washed 3 x for 10 min in 0.3% PBT (PBS with 0.3% Triton X-100, Sigma-Aldrich) and blocked for 30 min in 0.3% PBT supplemented with 5% normal goat serum (NGS). The preparations were incubated with primary antibodies (diluted in 0.3% PBT with 5% or 3% NGS) at 4°C overnight. After washing 6 x for 10 min with 0.3% PBT, the samples were incubated with secondary antibodies (diluted in 0.3% PBT with 5% or 3% NGS) for 120 min at RT or at 4°C overnight. Following 6 × 10 min washing steps with 0.3% PBT, the preparations were immersed in Vectashield (Vector Laboratories) and stored for at least one night at 4°C. The following primary antibodies were used: mouse-α-GFP (1:200; Sigma-Aldrich, SAB4200681; RRID:AB_2827519), rabbit-α-RFP (1:200; antibodies-online, ABIN129578; RRID:AB_10781500). Secondary antibodies: Alexa Fluor-488-conjugated goat-α-mouse (1:250; Invitrogen, A-11001; RRID:AB_2534069), Cy3-conjugated goat-α-rabbit (1:250; Jackson ImmunoResearch, 111-165-003; RRID:AB_2338006), Alexa Fluor-647 goat- α-rabbit (1:250; Invitrogen, A-21246; RRID:AB_2535814), and α-HRP conjugated with Alexa488 (1:250; Jackson ImmunoResearch, 123-545-021; RRID:AB_2338965). Samples were mounted in Vectashield and confocal images (*Figure 1*, *Figure 4—figure supplement 2*) were acquired with a LSM 800 (Zeiss) and a Leica TCS SP5. Intensity and contrast were set using Fiji (*Schindelin et al., 2012*) and Photoshop CC 2018 (Adobe).

## Nociceptor morphometry

For analyses of C4da neuron morphology, staged third instar larvae (96 ± 3 hr after egg laying, AEL), raised in density-controlled vials, were mounted in halocarbon oil. Confocal images of photoprotein signals (*ppk-GAL4 >UAS-CD4::tdGFP* or *UAS-CD4::tdTomato*) were collected with a Zeiss LSM700 laser scanning microscope. Image stacks with a Z step size between 0.5 and 2 μm were acquired from abdominal segment A5 with a 20×/0.8 objective and quantified with Imaris (Bitplane) using the filament tracer tool.

## Nocifensive behaviour

For mechanical nociception assays, wandering third instar larvae were collected in a sylgard covered Petri dish and stimulated with a 40 mN von Frey filament (made from fishing line, 0.22 mm diameter, Caperlan; calibrated with a precision balance). A single noxious mechanical stimulus was rapidly delivered to midabdominal segments (∼A4–A6) on the dorsal side of the larva. A positive response was scored if stimulation elicited at least one nocifensive corkscrew body roll. For all behavioural experiments each animal was tested only once. The adenylyl cyclase inhibitors SQ22536 and DDA were added to the food (giving final concentrations of 500 μM) one day prior to the experiments (*Figure 3D*). All data were collected from at least seven trials (N, *Table 1*) each sampling 6–53 larvae.

**Table 2.** 0: no response, 1: stimulated rolling, 2: spontaneous bending, 3: spontaneous rolling.

| Figure | Genotype | Mean 0 | Mean 1 | Mean 2 | Mean 3 | N |
|---|---|---|---|---|---|---|
| *Figure 3B* | wild-type (dark) | 47.29 | 52.71 | 0.00 | 0.00 | 203 |
| | wild-type (light) | 48.00 | 52.00 | 0.00 | 0.00 | 200 |
| | *ppk > bPAC* (dark) | 31.25 | 53.13 | 15.63 | 0.00 | 96 |
| | *ppk > bPAC* (light) | 5.13 | 56.41 | 28.21 | 10.26 | 39 |
| | *KO ppk > bPAC* (dark) | 0.00 | 70.00 | 26.67 | 3.33 | 30 |
| | *KO ppk > bPAC* (light) | 0.00 | 58.06 | 12.90 | 29.03 | 31 |

## Calcium imaging

Calcium imaging of C4da axon terminals was performed as previously described (*Figure 4—figure supplement 1*; *Hu et al., 2017*). Briefly, staged third instar larvae (96 ± 3 hr AEL) were pinned on a Sylgard (Dow Corning) plate and partially dissected in physiological saline to expose the ventral nerve cord (VNC). C4da neuron axon terminals expressing GCaMP6m were live-imaged by confocal microscopy with a 40 × water objective (Olympus FV1000MP) with 3x zoom to image at least four segments ensuring the calcium response could be detected. Activation of sensory neurons was achieved by providing a mechanonociceptive cue using a micromanipulator-mounted von Frey filament (45 mN) for stimulation of midabdominal segments (A3–A5). The most robust responses to local von Frey filament stimulation are restricted to a single VNC hemisegment corresponding to the stimulation site on the body wall although the adjacent segment(s) could also be slightly activated. The transient dip in fluorescence intensity (*Figure 4B*) is due to the larval brain moving out of focus briefly during and after mechanical stimulation. Baseline ($F_0$) and relative maximum intensity change ($\Delta F_{max}$) of GCaMP6m fluorescence were analysed.

## Optogenetics
### bPAC

Larvae were placed in a drop of water on a Sylgard-coated Petri dish and monitored with a stereomicroscope (Olympus SZX16). After applying blue light (~200 µW/mm$^2$ at 475 nm) for 3 min, the animals were mechanically stimulated with a von Frey filament (see above). Each larva was scored according to the following criteria. no response: no nocifensive or irregular behaviour during light or upon mechanical stimulation; stimulated rolling: no nocifensive or irregular behaviour during light, corkscrew body roll upon mechanical stimulation; spontaneous bending: head-swinging or bending during light; spontaneous rolling: corkscrew body roll during light. For each set of experiments, three larvae were analysed simultaneously and each animal was tested only once. In *Table 2*, N refers to the number of individuals tested.

**Table 3.** 0: no photoinduced response, 1: photoinduced response.

| Figure | Genotype | Mean 0 | Mean 1 | N |
|---|---|---|---|---|
| *Figure 4C* | wild-type ($10^{-3}$ mW/mm$^2$) | 100 | 0 | 20 |
| | wild-type ($10^{-2}$ mW/mm$^2$) | 100 | 0 | 20 |
| | wild-type ($10^{-1}$ mW/mm$^2$) | 100 | 0 | 20 |
| | *ppk > chop2$^{XXM}$* ($10^{-3}$ mW/mm$^2$) | 95 | 5 | 20 |
| | *ppk > chop2$^{XXM}$* ($10^{-2}$ mW/mm$^2$) | 25 | 75 | 20 |
| | *ppk > chop2$^{XXM}$* ($10^{-1}$ mW/mm$^2$) | 36.36 | 63.63 | 22 |
| | *dCirl$^{KO}$ppk > chop2$^{XXM}$* ($10^{-3}$ mW/mm$^2$) | 57.14 | 42.86 | 21 |
| | *dCirl$^{KO}$ppk > chop2$^{XXM}$* ($10^{-2}$ mW/mm$^2$) | 9.52 | 90.48 | 21 |
| | *dCirl$^{KO}$ppk > chop2$^{XXM}$* ($10^{-1}$ mW/mm$^2$) | 0 | 100 | 20 |

### ChR2$^{XXM}$

Larvae were placed in a Petri dish and illuminated with blue light (either ~$10^{-1}$ mW/mm$^2$, ~$10^{-2}$ mW/mm$^2$, or ~$10^{-3}$ mW/mm$^2$ at 475 nm) for 30 s. Each larva was scored according to the following criteria. No photoinduced response: no nocifensive or irregular behaviour during light exposure; photoinduced response: nocifensive responses such as massive head-swinging, bending, or cork-screw body roll during light exposure. For each experiment, two or three larvae were analysed simultaneously and each animal was tested only once. In *Table 3*, N refers to the number of individuals tested. Larvae were raised on food supplemented with 100 µM all-*trans*-retinal (Sigma-Aldrich).

### *Drosophila* neuropathy model

DMSO (dimethyl sulfoxide, vehicle, Sigma Aldrich) or paclitaxel (S1150 Absource Diagnostics, dissolved in DMSO) were carefully mixed into the food vials once ~90% of first instar larvae had hatched. This way, paclitaxel treatment (10 µM) occurred after completion of neurogenesis and axonal pathfinding (*Bhattacharya et al., 2012*).

## Rat experiments

### Traumatic neuropathy (Chronic constriction injury, CCI)

Animal care and protocols were performed in accordance with international guidelines for the care and use of laboratory animals (EU Directive 2010/63/EU for animal experiments) and were approved by the Government of Unterfranken (protocol numbers 2–733 and 2–264). Humane endpoints and criteria for discontinuation of the experiments with approved score sheets were defined, and the animals were treated accordingly. Animal studies were reported according to the ARRIVE guidelines (*McGrath and Lilley, 2015*). Male Wistar rats (200–250 g, Janvier labs, Le Genest-St-Isle, France) were housed in groups of six on dry litter (12 hr:12 hr light/dark cycle, 21–25°C, 45–55% humidity) with food and water ad libitum. All experiments were performed during the light phase and equal test groups (n = 6) were planned. Surgery of the Wistar rats was performed under deep isoflurane anaesthesia (1.8 Vol%, fiO2). After skin incision and exposure of the sciatic nerve by blunt preparation, four loose silk ligatures were made (Perma Silk 6.0, Ethicon Inc) with approximately 1 mm spacing in between (*Sauer et al., 2017*). After loosely tightening the ligatures, the skin was stitched (Prolene 5.0, Ethicon Inc). Animals were euthanized with $CO_2$ at the end of the experiment.

### Mechanical nociceptive thresholds

A series of von Frey filaments (Aesthesio set, Ugo Basile) were used to record the withdrawal threshold of the hind paw to identify the mechanical allodynia response (*Lux et al., 2019*) and touch sensitivity in neuropathy. Filaments were applied to the plantar surface of the hind paw and held for 1–3 s, until they were bent to a 45° angle. Each paw received stimuli with different filament forces, with a 30 s recovery period between each application. The 50% paw withdrawal threshold for von Frey filament responses was determined using Dixon's up and down method (*Dixon, 1980*).

### In situ hybridization and immunohistochemistry

After euthanizing the rats, DRGs were harvested, embedded in Tissue-Tek O.C.T. Compound (Sakura Finetek Europe B.V.), snap frozen in liquid nitrogen, and stored at −80°C. 10 µm thick cryosections were cut at −20°C (CM3050 S Research Cryostat, Leica Biosystems) and the slides were stored at −80°C until further use. For fixation, tissue sections were placed in precooled 4% PFA in DEPC (diethyl pyrocarbonate)-treated distilled water. Following washing steps with DEPC-treated reagents, probes for rat ADGRL1 and ADGRL3 (tagged with Cy3 and Cy5, respectively) were added for the RNAscope fluorescent multiplex assay (Advanced Cell Diagnostics, Inc), which was performed according to the manufacturer's instructions (document 320293-USM). After 15 min of incubation at 4°C the samples were dehydrated in ethanol at RT (50%, 70%, 100%, 100%; 5 min each). Hydrophobic barriers measuring approximately the same area, were drawn around the tissue sections and allowed to dry completely. Afterwards, each section was incubated with two drops of RNAscope Protease IV reagent (15 min at RT). Following the RNAscope assay, samples were washed, blocked with 10% donkey serum in PBS (1 hr at RT), and counterstained with neuronal markers (diluted in 10% donkey serum in PBS and added for two nights at 4°C). Non peptidergic nociceptors: isolectin B4 (IB4)-FITC conjugate (1:200, Sigma-Aldrich, L2895); large myelinated mechanosensors and

proprioceptors: rabbit anti-NF200 (1:200, Sigma-Aldrich, RRID:AB_477272); peptidergic nociceptors: mouse anti-CGRP (1:150, Abcam, RRID:AB_1658411). Following incubation, the sections were washed and secondary antibodies were added (diluted in PBS and incubated for 1 hr at RT). For CGRP: donkey anti-mouse AlexaFluor488 (1:1000, Life Technologies, RRID:AB_141607), for NF200: donkey anti-rabbit AlexaFluor488 (1:1000, Life Technologies, RRID:AB_141708). After washing, the slides were mounted in Vectashield, dried for 15 min at RT, and stored at 4°C until imaging (<24 hr). All images were acquired in one session by confocal microscopy (Olympus FV1000) with a 20×/0.75 objective (Olympus UPlan SAPO) using the same parameters for z-stacks (1 µm step size) of Cy3 and Cy5 channels.

## Image evaluation

Images were processed with Fiji (*Schindelin et al., 2012*) by scientists blinded to the test groups. The markers of neuronal subpopulations were used to identify complete and distinguishable cells as regions of interest (ROIs) for further analysis. Thresholds were applied to maximal projections of confocal z-stacks for Cy3 (grey value 1200) and Cy5 (grey value 1150) signals (this procedure was also used for the example images in *Figure 6D,E*). Elements between 3 and 22 pixels in size, defined as mRNA clusters, were quantified for each channel following background subtraction. An independent approach based on computational image evaluation gave comparable results (data not shown). Here, a convolutional neural network (DeepFLaSh) (*Segebarth et al., 2020*) was trained with six images for each neuronal marker (NF200, IB4, CGRP) and then used to identify Cy3 and Cy5 clusters.

## Statistics

Data were analysed with Prism 8.2 (GraphPad). Group means were compared by an unpaired two-tailed t-test, unless the assumption of normal sample distribution was violated, in which case group means were compared by a nonparametric Mann–Whitney rank sum test. To compare more than two groups an ordinary one-way ANOVA with Tukey correction (normal distribution) or a Kruskal–Wallis test (not normally distributed) were used.

## Acknowledgements

We thank PA Stevenson, I Maiellaro, and J Eilers for discussions, R Blum for help with the computational image analysis of DRGs, and N Naumann and B Goettgens for technical assistance. This work was supported by the Graduate School of Life Sciences, University of Würzburg and grants from the German Research Foundation (DFG) to MS (PA3241/2-1), HLR (RI817/13-1), RJK (FOR 2149/P03, TRR 166/B04, KI1460/4-1, KI1460/5-1), PS (SPP 1926/SO1337/2-2, SO1337/4-1), DP (PA1979/2-1, PA1979/3-1), and TL ( FOR 2149/P01 and P03, CRC 1423, project number 421152132, subproject B06). Stocks obtained from the Bloomington *Drosophila* Stock Center (NIH P40OD018537) and the Vienna *Drosophila* Resource Center (*Dietzl et al., 2007*) were used in this study. The authors declare no competing financial interests.

## Additional information

### Funding

| Funder | Grant reference number | Author |
| --- | --- | --- |
| Deutsche Forschungsgemeinschaft | PA3241/2-1 | Mareike Selcho |
| Deutsche Forschungsgemeinschaft | RI817/13-1 | Heike L Rittner |
| Deutsche Forschungsgemeinschaft | FOR 2149/P03 | Robert J Kittel |
| Deutsche Forschungsgemeinschaft | TRR 166/B04 | Robert J Kittel |
| Deutsche Forschungsgemeinschaft | KI1460/4-1 | Robert J Kittel |

meinschaft

| | | |
|---|---|---|
| Deutsche Forschungsge-meinschaft | KI1460/5-1 | Robert J Kittel |
| Deutsche Forschungsge-meinschaft | SPP 1926/SO1337/2-2 | Peter Soba |
| Deutsche Forschungsge-meinschaft | SO1337/4-1 | Peter Soba |
| Deutsche Forschungsge-meinschaft | FOR 2149/P01 and P03 | Tobias Langenhan |
| Deutsche Forschungsge-meinschaft | PA1979/2-1 | Dennis Pauls |
| Deutsche Forschungsge-meinschaft | PA1979/3-1 | Dennis Pauls |
| Deutsche Forschungsge-meinschaft | CRC 1423/B06 | Tobias Langenhan |

The funders had no role in study design, data collection and interpretation, or the decision to submit the work for publication.

### Author contributions

Sven Dannhäuser, Thomas J Lux, Chun Hu, Jeremy T-C Chen, Nadine Ehmann, Divya Sachidanandan, Formal analysis, Investigation, Writing - review and editing; Mareike Selcho, Formal analysis, Supervision, Funding acquisition, Investigation, Writing - review and editing; Sarah Stopp, Investigation, Writing - review and editing; Dennis Pauls, Investigation, Writing - review and editing, Funding acquisition; Matthias Pawlak, Conceptualization, Formal analysis, Writing - review and editing; Tobias Langenhan, Conceptualization, Supervision, Funding acquisition, Writing - review and editing; Peter Soba, Heike L Rittner, Conceptualization, Formal analysis, Supervision, Funding acquisition, Writing - review and editing; Robert J Kittel, Conceptualization, Formal analysis, Supervision, Funding acquisition, Investigation, Writing - original draft, Project administration, Writing - review and editing

### Author ORCIDs

Thomas J Lux http://orcid.org/0000-0003-1049-9872
Divya Sachidanandan http://orcid.org/0000-0001-8219-8177
Tobias Langenhan http://orcid.org/0000-0002-9061-3809
Heike L Rittner https://orcid.org/0000-0003-4867-0188
Robert J Kittel https://orcid.org/0000-0002-9199-4826

### Ethics

Animal experimentation: Animal care and protocols were performed in accordance with international guidelines for the care and use of laboratory animals (EU Directive 2010/63/EU for animal experiments) and were approved by the Government of Unterfranken (protocol numbers 2-733 and 2-264).

### Decision letter and Author response

Decision letter https://doi.org/10.7554/eLife.56738.sa1
Author response https://doi.org/10.7554/eLife.56738.sa2

## Additional files

### Supplementary files

• Transparent reporting form

### Data availability

The presented data are summarized in Tables 1-3.

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
