## [Decision Letter]

**Acceptance summary:**

You provide compelling evidence that aGPCR is expressed in high-threshold mechanosensory neurons where it regulates nocifensive (pain-like) behaviors. The systemic ablation of *cirl*, or its specific knockdown in nociceptive neurons, leads to an enhanced nocifensive response, whereas overexpression of the gene elicits an attenuated response. You also provide evidence that vertebrate chronic pain models are associated with diminished expression of the *cirl* homologs, indicating a potential role in the onset of chronic and pathological pain in vertebrates. Based on these observations the three reviewers agree that your manuscript is appropriate for *eLife*.

**Decision letter after peer review:**

[Editors’ note: the authors submitted for reconsideration following the decision after peer review. What follows is the decision letter after the first round of review.]

Thank you for choosing to send your work, "Antinociceptive modulation by the adhesion-GPCR CIRL promotes mechanosensory signal discrimination", for consideration at *eLife*. Your article has been reviewed by three peer reviewers, and the evaluation has been overseen by a Reviewing Editor and a Senior Editor. Although the work is of interest, we regret to inform you that the findings at this stage are too preliminary for consideration at *eLife* at this time.

The reviewers felt that the data are interesting but have raised many concerns that will require additional experiments. We realize that at this time, during COVID, most labs are not open and you will not be able to do these experiments. For these reasons, we will give you the option of either going to a different journal entirely, or coming back to *eLife* with a substantially revised version at some point in the future. If you come back to *eLife*, we would consider that a new submission but would do everything we can to engage the same reviewers for this process.

We do not intend any criticism of the quality of the data or the rigor of the science. We wish you good luck with your work and we hope you will consider *eLife* for future submissions.

Reviewer #1:

In a study that is a follow-up of the original characterization of CIRL in *Drosophila* mechanotransduction, Dannhäuser et al. demonstrate that the aGPCR is also expressed in high-threshold mechanosensory neurons where it regulates nocifensive (pain-like) behaviors. The authors demonstrate that owing to its coupling with Gi, CIRL lowers cAMP levels in a cell autonomous manner. Either the systemic ablation of *cirl*, or its specific knockdown in nociceptive neurons, led to an enhanced nocifensive response, whereas overexpression of the gene in those same neurons elicited an attenuated response. The authors claim CIRL influences the overall activity of the nociceptive neurons, an argument based on Ca^2+^ imaging of nociceptive neurons. Finally, the authors provide experimental evidence that vertebrate chronic pain models are associated with diminished expression of the *cirl* homologs, which suggests a potential role in in the onset of chronic and pathological pain in vertebrates.

On the whole the manuscript is well written and makes good use of *Drosophila* as a model to examine the neurogenetics of nociception. There are, however, some concerns that diminish enthusiasm and need to be addressed before publication in *ELife*.

1) The authors make no reference to the ligand that is needed for the activation of CIRL in the context of nocifensive behavior. They write that CIRL is activated by a tethered protein ligand encoded by Strachel. Have they not examined whether deletion or knockdown of Strachel leads to similar nocifensive phenotypes? Given that ligand-induced activation of GPCRs is central to our understanding of biological function of those receptors, identification of ligand is critical to the proposed model. This reviewer appreciates the difficulties associated with de novo identification of GPCR ligands, and is not asking for a full examination. However, the role for Strachel in nocifensive behavior can and should be examined.

2) In the Ca^2+^ imaging data shown in Figure 4 it appears that the authors are imaging regions of the larval brain that harbor the axon termini of nociceptive neurons. They argue that the response amplitudes are larger in following *cirl* knockdown. These data are not compelling for several reasons. First, the observed Ca^2+^ responses are probably a function of the pressure applied to the periphery of the animals. Given the involvement of second messengers that typically amplify signals, minute variations in the pressure of the initial 'poke' could elicit Ca^2+^ responses of wildly different amplitudes. How can the authors know that the location and pressure of the stimulus is comparable between different data points? Second, control animals show increased GCaMP signals in segments A3 and A4, whereas the mutant animals show responses in A4 and A5. Which segments were stimulated in these recordings, and did the responses correlate with the segments that were stimulated? Third, the data should be presented as Ca^2+^ traces (amplitudes vs. time) so that readers can evaluate the temporal relationships between stimulation and Ca^2+^ elevation. Fourth, the authors should indicate the frequency of spontaneous activity at the axon termini.

3) The authors demonstrate that in mice with sciatic ligatures, expression of the homologs of *cirl* decreases. This is a very interesting finding that could be of broad interest to the field of chronic pain. Can the authors provide any evidence that the knockdown of *cirl* homologs in DRG neurons is associated with an increase in cAMP levels or cAMP-dependent signalling? Is the knockdown of *cirl* homologs in DRG neurons sufficient to trigger hyperexcitability of elevated Ca^2+^ responses?

4) A minor textual issue relates to the authors' statement that CIRL modulates the processing of innocuous and noxious stimuli a bidirectional manner. In the opinion of this reviewer, this statement is not fully accurate. In both populations of neurons, CIRL couples to Gi such that engagement of the agonist leads to a decrease in cAMP levels. The molecular function of CIRL – lowering cAMP levels – is the same in both populations. The difference in behavior is a function of the neuron type under consideration. In one population, lowering cAMP leads to increased activity whereas the same signal correlates with decreased activity in the other population. If anything, it is cAMP that modulates the processing of innocuous and noxious stimuli a bidirectional manner. Any receptor that mimics CIRL in the regulation of cAMP in those neurons would elicit the similar behaviors. If the authors agree with this notion, they should consider rewording the relevant portions of the manuscript.

5) Another minor comment relates to the use of the term, "quenches." Gi activation decreases the activity of AC leading to a decrease in the synthesis of cAMP. This is not equivalent to a "quench", a term that implies the breakdown of cAMP molecules already present in the cell. It is recommended that the authors change the text suitably.

Reviewer #2:

This study by Dannhauser et al. investigates the function of the adhesion GPCR *Cirl* in modulating nociceptive signalling in pain-sensing neurons in the *Drosophila* periphery. Building from a previous study published in *eLife* in which the authors demonstrated that *Cirl* worked to sensitize sensory responses in low-threshold mechanosensory neurons, this group now characterizes the function of *Cirl* in nociceptive neurons. As such, this study is well suited as a Research Advance. Using a combination of genetic manipulations, behavioral assays, and functional imaging, the authors show that *Cirl* essentially functions similarly in nociceptive neurons as it does in mechanosensory neurons, working to reduce cAMP levels. However, the downstream signal transduction machinery in nociceptive neurons apparently interprets the *Cirl*-dependent reduction in cAMP levels differently, ultimately diminishing the response of nociceptors to mechanical stimulation. Rather than *Cirl* itself functioning differently between touch and pain neurons, it appears that it is the downstream cAMP signal transduction machinery that operates in different ways.

The demonstration that the same signal transduction molecules can be interpreted differently by distinct cell types is not particularly surprising, as this has been well understood for decades in diverse fields (developmental patterning, axon guidance, etc). Importantly, *Cirl* levels are capable of bi-directionally tuning the sensitivity of nociceptive neurons, and the optogenetic experiments manipulating cAMP levels along with the calcium imaging serve to link *Cirl*-dependent signalling to neuronal output. Overall, this is an interesting study, the manuscript is well written and nicely sets up the framework and significance of the findings. What is missing is an understanding of to what extent CIRL is a target for physiological modulation of nociception in *Drosophila* – i.e., in what context and how would *Cirl*-dependent signalling adaptively (or maladaptively) calibrate the sensitivity of mechanosensation in touch and/or nociception? Similar to the previous study, it is also not clear how cAMP levels are transformed to specific changes in mechanosensitivity. Given the understandable efforts to avoid imposing additional experiments during this time of covid-19 precautions, I would suggest the following to improve the manuscript.

1) Physiological signals that regulate CIRL-mediated modulation of nociception: The rat data shown in Figure 6 is used by the authors to argue that *Cirl1* may function in mammals similarly to *Drosophila*: "these correlative results are consistent with the *Drosophila* data linking low *Cirl* expression levels to nociceptor sensitization": This language is problematic, as the authors have not shown that downregulation of *Cirl* expression is an actual mechanism used in vivo in *Drosophila* to sensitize nociception. Rather, the authors have deliberately manipulated *Cirl* expression and observed sensitization. This holds true for the experiments in Figure 5 using the paclitaxel assay as well. However, in the rat experiment, the authors seem to imply an endogenous signalling system that ultimately served to reduce *Cirl* expression in response to chronic activation of neuropathic pain. The authors should make clear these distinctions and the lack of evidence from their *Drosophila* studies that *Cirl* activity is directly modulated to calibrate pain sensitivity.

The authors should also show the primary in situ hybridization images used for quantification in Figure 6 so readers can assess the data.

2) Re-framing the understanding of *Cirl* and cAMP signalling: Throughout this study, the authors really demonstrate that *Cirl* functions similarly in nociceptors and low-threshold mechanosensitive neurons – in both neuronal subtypes, *Cirl* functions to reduce cAMP signalling. Ultimately, is it simply the signal transduction downstream of cAMP that is unique to the two cell types, not *Cirl* itself. The authors do a good job of suggesting that the distinctions may be in the separate mechanosensitive proteins expressed in the neuronal subtypes in the Discussion. But at minimum the authors should make it clear that 1) *Cirl* functions similarly in both neurons to reduce cAMP signalling and 2) there is no evidence yet that *Cirl*-related signalling actually changes to adapt or tune mechanosensitivity in *Drosophila*.

Reviewer #3:

The authors examine the role of an adhesion GPCR *dCirl* in the function of the nociceptive neurons of the *Drosophila* larva. They show 1) that a reporter for *dCirl* is expressed in the cIVda nociceptors, 2) mutants for *dCirl* show hypersensitive mechanical nociception behaviors and the reverse is seen with *dCirl* over expression. 3) There is elevated Ca in the axon terminals is observed upon mechanical stimulation. 4) increasing cAMP in the neurons via an optogenetic sensitizes nociception responses 5) manipulating *dCirl* levels affects nociceptor branching 6) *dCirl* over expression reduces paclitaxel induced hyperalgesia 7) mammalian Cirl genes are down regulated in rats with sciatic nerve ligation injury. Overall, the data appear to be quite robust and the authors do make a convincing case that *dCirl* negatively regulates the sensitivity of the nociceptive neurons. Nevertheless, the paper is largely descriptive and provides little mechanistic understanding of how the phenotype actually occurs. Neither the mechanisms of how *dCirl* may be activated in the nociceptors nor the downstream effectors of its action have been determined. The rat experiments seem gratuitously added on and the paclitaxel experiments also provide little insight.

The authors find that *dCirl* negatively regulates nociceptor excitability and their prior studies found that *dCirl* has the opposite effect in chordotonal organs. Based on this they claim that they report "a new molecular principle underlying the processing of mechanical input." This is a vastly overstated claim as it is extremely well understood that same GPCR can have opposing effects in different contexts depending on the downstream effectors. Most GPCRs can be coupled to various G α subunits and therefore can have variable effects in different cell-types. As well, signalling can occur through β-arrestin, GRKs, Srcs etc… the present study does little to address the mechanism of the opposing effects of *dCirl* in Cho's vs. cIVda neurons.

*dCirl^KO^* enhances the effects of increasing cAMP in the neurons may indicate *dCirl* loss is increasing cAMP but it could also indicate a parallel pathway. Epistasis experiments of the *dCirl^KO^* are needed to resolve this issue.The authors really should investigate the effects of *dCirl* on thermal nociception and optogenetic triggered to determine if the effect is specific to mechanical nociception or to excitability of the cells in general.

Prior literature in the field that has implicated GPCR signalling in the *Drosophila* nociception pathways needs to be cited and incorporated into the Discussion:

Kaneko et al., 2017; Herman et al., 2018; Honjo et al., 2016 (found G α o as a negative regulator of thermal nociception); Christianson, Mauthner and Tracey, 2016 (found G α o as a negative regulator of mechanical nociception).

---

## [Author Response]

[Editors’ note: The authors appealed the original decision. What follows is the authors’ response to the first round of review.]

Reviewer #1:[…] On the whole the manuscript is well written and makes good use of *Drosophila* as a model to examine the neurogenetics of nociception. There are, however, some concerns that diminish enthusiasm and need to be addressed before publication in eLife.1) The authors make no reference to the ligand that is needed for the activation of CIRL in the context of nocifensive behavior. They write that CIRL is activated by a tethered protein ligand encoded by Strachel. Have they not examined whether deletion or knockdown of Strachel leads to similar nocifensive phenotypes? Given that ligand-induced activation of GPCRs is central to our understanding of biological function of those receptors, identification of ligand is critical to the proposed model. This reviewer appreciates the difficulties associated with de novo identification of GPCR ligands, and is not asking for a full examination. However, the role for Strachel in nocifensive behavior can and should be examined.

We would like to thank the reviewer for pointing out that the role of *Cirl’s* tethered agonist should be examined in more detail. The additional experiments have added an important new aspect to the manuscript.

Several studies have shown that aGPCR activation can occur by means of an intramolecular tethered agonist, termed *Stachel* (stalk; Liebscher et al., 2014; Monk et al., 2015; Stoveken et al., 2015). This linker sequence of approximately 20 amino acids begins at the extracellular GPCR proteolysis site (GPS) and connects the GAIN domain to the 7-transmembrane unit. In the revised manuscript, we now illustrate this layout in Figure 4D.

In line with tethered agonism, our previous study in *eLife*, which this Research Advance builds upon, showed that CIRL function requires an intact *Stachel* in touch-sensitive ChO neurons (Scholz et al., 2017). Notably, though, mutating the *Stachel* sequence of CIRL3 does not disrupt receptor function in the mouse hippocampus (Sando et al., 2019). Moreover, work in cell culture has put forward a model in which *Stachel-*dependent and -independent aGPCR activation may co-exist and trigger different downstream signalling pathways (Kishore et al., 2016; Salzman et al., 2017).

In order to test for a role of the *Stachel* sequence in activating CIRL in mechanical nociceptors, we now used two established mutants. Whereas the *dCirl^T>A^* allele carries a mutation at the +1 position of the GPS within the *Stachel* sequence, the -2 mutation in the *dCirl^H>A^* allele leaves the *Stachel* intact but reduces protein expression (Scholz et al., 2017). Analysing nocifensive behaviour in these mutants shows normal responses for the T>A allele and a *Cirl* null-mutant phenocopy for the H>A allele (Figure 4E). Thus, the nociceptors appear to be sensitive to reduced CIRL expression, which is consistent with very low protein levels in wild-type (close to the detection threshold; see new Figure 4—figure supplement 2). An intact tethered agonist, in turn, is dispensable for CIRL function in nociceptive neurons, contrasting the situation in touch-sensitive ChO neurons, where mutating the *Stachel* sequence disrupts receptor function.

This highly interesting finding argues that the mechanism of CIRL activation differs in these two types of mechanosensory neurons and provides evidence in support of a dual activation model of an aGPCR in a physiological setting. Moreover, the new results raise the possibility that this functional differentiation may be connected to specific downstream effects (see also reviewer #3 response 4).

The new results are described and discussed in detail in the revised manuscript (Introduction, last paragraph; Results, sixth paragraph; Discussion, third paragraph).

2) In the Ca^2+^ imaging data shown in Figure 4 it appears that the authors are imaging regions of the larval brain that harbor the axon termini of nociceptive neurons. They argue that the response amplitudes are larger in following cirl knockdown. These data are not compelling for several reasons. First, the observed Ca^2+^ responses are probably a function of the pressure applied to the periphery of the animals. Given the involvement of second messengers that typically amplify signals, minute variations in the pressure of the initial 'poke' could elicit Ca^2+^ responses of wildly different amplitudes. How can the authors know that the location and pressure of the stimulus is comparable between different data points? Second, control animals show increased GCaMP signals in segments A3 and A4, whereas the mutant animals show responses in A4 and A5. Which segments were stimulated in these recordings, and did the responses correlate with the segments that were stimulated? Third, the data should be presented as Ca^2+^ traces (amplitudes vs. time) so that readers can evaluate the temporal relationships between stimulation and Ca^2+^ elevation. Fourth, the authors should indicate the frequency of spontaneous activity at the axon termini.

We thank the reviewer for this comment. For these experiments, we use semi-intact preparations (opening segment A1-A2 to expose the ventral nerve cord while keeping A2/A3-A9 segments intact), leaving the posterior larval body wall intact but exposing the larval brain (see new Figure 4—figure supplement 1). The stimulus is then applied to an abdominal segment (segment A3-A5 as spelled out in the Materials and methods section) using a calibrated von Frey filament (45mN) mounted on a micromanipulator. Typically, the filament will only elicit responses in 1 segment (always targeting A3-A5), which means only the poked segment showed the axonal calcium response. Stimulation force is determined mainly by the filament strength, which bends upon reaching the maximum force (45 mN). In addition, micromanipulator movement speed is kept constant for stimulation. However, the exact placement of the filament on a body wall segment can only be reconciled by correlating it with the corresponding calcium responses in the nociceptor axon terminals, as imaging (brain) and stimulation (posterior body wall) occur in different regions of the animal. This established and published method (Hu et al., 2017) faithfully reports calcium responses in axon terminals of nociceptors, and has been successfully used to report differences due to genetic perturbations as shown here as well. Responses in the targeted segments (A3-A5) are indistinguishable due to the presence of the same number and type of nociceptors in all segments, and repeated stimulation gives rise to highly comparable responses. Equally, stimulation applied to A3-A5 segments of intact larvae gives rise to identical behavioral escape responses with comparable frequency.

The examples originally shown for the two conditions differ in terms of the targeted segment, yet they are still representative for the entire dataset and fully comparable to previous data obtained under the same conditions (Hu et al., 2017). Nonetheless, we have now chosen two examples showing maximum stimulation in the same segment (new Figure 4A). In all cases, we quantified the signal in the segment showing the highest response, as only one body wall segment is strongly stimulated by the von Frey filament. We however agree with the reviewer that calcium traces allow a more objective evaluation of the data, which we now provide in Figure 4B. Please note that due to the mechanical stimulation and corresponding body wall movement, the larval brain is briefly out of focus during and after stimulation (recognizable by the characteristic dip in fluorescence intensity). Of note, the extracted maxima used for the F_max_ plot (Figure 4B) are from the same data as the calcium traces, yet the F_max_ values appear larger as the maximum fluorescence change in each sample is not occurring at precisely the same time, thus reducing the mean fluorescence change over time.

Lastly, the sensitivity of the GCaMP6m reporter under these conditions does not allow us to visualize spontaneous activity in the axon termini. Baseline fluorescence signals show very little fluctuation and detectable calcium responses as shown here occur only after stimulation (<0.5 s), i.e. they are tightly linked to the mechanical stimulus and never observed without stimulation.

3) The authors demonstrate that in mice with sciatic ligatures, expression of the homologs of cirl decreases. This is a very interesting finding that could be of broad interest to the field of chronic pain. Can the authors provide any evidence that the knockdown of cirl homologs in DRG neurons is associated with an increase in cAMP levels or cAMP-dependent signalling? Is the knockdown of cirl homologs in DRG neurons sufficient to trigger hyperexcitability of elevated Ca^2+^ responses?

Answering these questions will require RNAi targeting of *Cirl1* and *Cirl3* in rat DRG neurons and/or knockout mice. For the mutant studies, we will likely need to use nociceptor-specific knockouts, which in turn will necessitate extended crossing and breeding. Both approaches will require new IRB approvals. We fully agree that the suggested experiments are of interest and should be carried out in detail in future studies. As a Research Advance, however, we feel that this work is beyond the scope of the present manuscript.

4) A minor textual issue relates to the authors' statement that CIRL modulates the processing of innocuous and noxious stimuli a bidirectional manner. In the opinion of this reviewer, this statement is not fully accurate. In both populations of neurons, CIRL couples to Gi such that engagement of the agonist leads to a decrease in cAMP levels. The molecular function of CIRL – lowering cAMP levels – is the same in both populations. The difference in behavior is a function of the neuron type under consideration. In one population, lowering cAMP leads to increased activity whereas the same signal correlates with decreased activity in the other population. If anything, it is cAMP that modulates the processing of innocuous and noxious stimuli a bidirectional manner. Any receptor that mimics CIRL in the regulation of cAMP in those neurons would elicit the similar behaviors. If the authors agree with this notion, they should consider rewording the relevant portions of the manuscript.

We have now taken care to explain more clearly how CIRL exerts opposing/bidirectional effects on the activity of neurons that process innocuous vs. noxious stimuli. That is, while CIRL lowers cAMP in both neuron types, the effect of cAMP on cellular activity differs in ChO and C4da neurons. In addition, to rewording the text (Abstract; Introduction, last paragraph; Discussion, first and last paragraphs; Figure 7 legend) we have modified Figure 7 to illustrate this principle.

5) Another minor comment relates to the use of the term, "quenches." Gi activation decreases the activity of AC leading to a decrease in the synthesis of cAMP. This is not equivalent to a "quench", a term that implies the breakdown of cAMP molecules already present in the cell. It is recommended that the authors change the text suitably.

“quench” has now been replaced throughout the text.

Reviewer #2:[…] Overall, this is an interesting study, the manuscript is well written and nicely sets up the framework and significance of the findings. What is missing is an understanding of to what extent CIRL is a target for physiological modulation of nociception in *Drosophila* – i.e., in what context and how would Cirl-dependent signalling adaptively (or maladaptively) calibrate the sensitivity of mechanosensation in touch and/or nociception? Similar to the previous study, it is also not clear how cAMP levels are transformed to specific changes in mechanosensitivity. Given the understandable efforts to avoid imposing additional experiments during this time of covid-19 precautions, I would suggest the following to improve the manuscript.1) Physiological signals that regulate CIRL-mediated modulation of nociception: The rat data shown in Figure 6 is used by the authors to argue that Cirl1 may function in mammals similarly to *Drosophila*: "these correlative results are consistent with the *Drosophila* data linking low Cirl expression levels to nociceptor sensitization": This language is problematic, as the authors have not shown that downregulation of Cirl expression is an actual mechanism used in vivo in Drosophila to sensitize nociception. Rather, the authors have deliberately manipulated Cirl expression and observed sensitization. This holds true for the experiments in Figure 5 using the paclitaxel assay as well. However, in the rat experiment, the authors seem to imply an endogenous signalling system that ultimately served to reduce Cirl expression in response to chronic activation of neuropathic pain. The authors should make clear these distinctions and the lack of evidence from their *Drosophila* studies that Cirl activity is directly modulated to calibrate pain sensitivity.

We thank the reviewer for the thoughtful comments and for pointing out this important consideration. In the revised manuscript, we have clarified that “It remains to be determined whether expression of *DrosophilaCirl* can be regulated physiologically to adapt or tune nociceptor sensitivity and whether activation of mammalian CIRL, in turn, can provide analgesia.”

The authors should also show the primary in situ hybridization images used for quantification in Figure 6 so readers can assess the data.

Example images are now shown in Figure 6D and E.

2) Re-framing the understanding of Cirl and cAMP signalling: Throughout this study, the authors really demonstrate that Cirl functions similarly in nociceptors and low-threshold mechanosensitive neurons – in both neuronal subtypes, Cirl functions to reduce cAMP signalling. Ultimately, is it simply the signal transduction downstream of cAMP that is unique to the two cell types, not Cirl itself. The authors do a good job of suggesting that the distinctions may be in the separate mechanosensitive proteins expressed in the neuronal subtypes in the Discussion. But at minimum the authors should make it clear that 1) Cirl functions similarly in both neurons to reduce cAMP signalling and 2) there is no evidence yet that Cirl-related signalling actually changes to adapt or tune mechanosensitivity in *Drosophila*.

1) We agree that this signalling principle should be made quite clear. Indeed, this point was also raised by reviewer #1 (see response 4). We have now made appropriate changes to the text (Abstract; Introduction, last paragraph; Discussion, first and last paragraphs; Figure 7 legend) and have included additional illustrations in Figure 7.

2) This clarification has now been added to the text (Results, last paragraph).

Reviewer #3:The authors examine the role of an adhesion GPCR dCirl in the function of the nociceptive neurons of the *Drosophila* larva. They show 1) that a reporter for dCirl is expressed in the cIVda nociceptors, 2) mutants for dCirl show hypersensitive mechanical nociception behaviors and the reverse is seen with dCirl over expression. 3) There is elevated Ca in the axon terminals is observed upon mechanical stimulation. 4) increasing cAMP in the neurons via an optogenetic sensitizes nociception responses 5) manipulating dCirl levels affects nociceptor branching 6) dCirl over expression reduces paclitaxel induced hyperalgesia 7) mammalian Cirl genes are down regulated in rats with sciatic nerve ligation injury. Overall, the data appear to be quite robust and the authors do make a convincing case that dCirl negatively regulates the sensitivity of the nociceptive neurons. Nevertheless, the paper is largely descriptive and provides little mechanistic understanding of how the phenotype actually occurs. Neither the mechanisms of how dCirl may be activated in the nociceptors nor the downstream effectors of its action have been determined. The rat experiments seem gratuitously added on and the paclitaxel experiments also provide little insight.The authors find that dCirl negatively regulates nociceptor excitability and their prior studies found that dCirl has the opposite effect in chordotonal organs. Based on this they claim that they report "a new molecular principle underlying the processing of mechanical input." This is a vastly overstated claim as it is extremely well understood that same GPCR can have opposing effects in different contexts depending on the downstream effectors. Most GPCRs can be coupled to various G α subunits and therefore can have variable effects in different cell-types. As well, signalling can occur through β-arrestin, GRKs, Srcs etc… the present study does little to address the mechanism of the opposing effects of dCirl in Cho's vs. cIVda neurons.

The cited statement has been removed. The new results on CIRL activation, i.e. *Stachel*-dependent vs. -independent (see reviewer #1 response 1, Introduction, last paragraph; Results, sixth paragraph, Figure 4D, E), and downstream signalling, i.e. enhancing mechanotransduction vs. reducing cellular excitability (see below), now provide further details on the opposing effects of CIRL in ChOs vs. C4da neurons (Figure 7, Discussion, third paragraph).

dCirl^KO^ enhances the effects of increasing cAMP in the neurons may indicate dCirl loss is increasing cAMP but it could also indicate a parallel pathway. Epistasis experiments of the dCirl^KO^ are needed to resolve this issue.

We thank the reviewer for pointing out this critical experiment. To substantiate that CIRL influences nociception by acting on cAMP-dependent signalling, we inhibited the endogenous adenylyl cyclase activity by pharmacological means. Consistent with nociceptor sensitization by cAMP, dietary supplementation with the adenylyl cyclase inhibitors SQ22536 or DDA significantly decreased nocifensive behaviour of *Drosophila* larvae. Importantly, this treatment produced indistinguishable responses of control and *dCirl^KO^* animals to von Frey filament stimulation. This result shows that CIRL acts in the same pathway as cAMP production by the adenylyl cyclase and supports the notion that the aGPCR exerts its antinociceptive effect by lowering cAMP levels through G_i/o_ -mediated inhibition of adenylyl cyclase activity. These new data are presented in Figure 3D and are included in the fourth paragraph of the Results, in the third paragraph of the Discussion and in the Materials and methods subsection “Nocifensive behaviour”.

The authors really should investigate the effects of dCirl on thermal nociception and optogenetic triggered to determine if the effect is specific to mechanical nociception or to excitability of the cells in general.

We fully agree, this is a valid point. In principle, CIRL could influence the activity of C4da neurons by modulating cellular excitability or by modulating the mechanotransduction process, as is the case in touch-sensitive ChO neurons (Scholz et al., 2017). To differentiate between these possibilities, we chose the optogenetic strategy and circumvented mechanotransduction by stimulating the nociceptors with light. Photostimulation of C4da neurons via the optimized Channelrhodpsin-2 (ChR2) variant ChR2^XXM^ triggered nocifensive behaviour, consistent with previous work (Hwang et al., 2007). Notably, *dCirl^KO^* larvae (*dCirl^KO^ppk-GAL4 > UAS-chop2^XXM^*) responded more strongly than controls (*ppk-GAL4 > UAS-chop2^XXM^*) over a range of different light intensities. This demonstrates that CIRL decreases the excitability of mechanical nociceptors, contrasting its role in ChO neurons where the aGPCR specifically enhances mechanotransduction. These results are shown in Figure 4C (now titled “*Cirl* decreases the excitability of nociceptors”) and are included in the fifth paragraph of the Results, the third paragraph of the Discussion and in the Materials and methods subsection “ChR2^XXM^”.

Prior literature in the field that has implicated GPCR signalling in the *Drosophila* nociception pathways needs to be cited and incorporated into the Discussion:Kaneko et al., 2017; Herman et al., 2018; Honjo et al., 2016 (found G α o as a negative regulator of thermal nociception); Christianson, Mauthner and Tracey, 2016 (found G α o as a negative regulator of mechanical nociception).

The recommended literature has now been cited in the first paragraph of the Discussion.